# FUNCTION CONTRASTIVE LEARNING OF TRANSFERABLE REPRESENTATIONS

## ABSTRACT

Few-shot-learning seeks to find models that are capable of fast-adaptation to novel tasks which are not encountered during training. Unlike typical few-shot learning algorithms, we propose a contrastive learning method which is not trained to solve a set of tasks, but rather attempts to find a good representation of the underlying data-generating processes (*functions*). This allows for finding representations which are useful for an entire series of tasks sharing the same function. In particular, our training scheme is driven by the self-supervision signal indicating whether two sets of samples stem from the same underlying function. Our experiments on a number of synthetic and real-world datasets show that the representations we obtain can outperform strong baselines in terms of downstream performance and noise robustness, even when these baselines are trained in an end-to-end manner.

## 1 INTRODUCTION

The ability to learn new concepts from only a few examples is a salient characteristic of intelligent behaviour. Nevertheless, contemporary machine learning models consume copious amounts of data to learn even seemingly basic concepts. The mitigation of this issue is the ambition of the few-shot learning framework, wherein a fundamental objective is to learn representations that apply to a variety of different problems (Bengio et al., 2019). In this work, we propose a self-supervised method for learning such representations by leveraging the framework of contrastive learning.

We consider a setting very similar to the one in Neural Processes (NPs) (Garnelo et al., 2018a;b; Kim et al., 2019): The goal is to solve some task related to an unknown function $f$ after observing just a few input-output examples $O^f = \{(x_i, y_i)\}_i$. For instance, the task may consist of predicting the function value $y$ at unseen locations $x$, or it may be to classify images after observing only a few pixels (in that case $x$ is the pixel location and $y$ is the pixel value). To solve such a task, the example dataset $O^f$ needs to be encoded into some representation of the underlying function $f$. Finding a good representation of a function which facilitates solving a wide range of tasks, sharing the same function, is the object of the present paper.

Most existing methods approach this problem by optimizing representations in terms of reconstruction i.e., prediction of function values $y$ at unseen locations $x$ (see e.g. NPs (Garnelo et al., 2018a;b; Kim et al., 2019) and Generative Query Networks (GQNs) (Eslami et al., 2018)). A problem with this objective is that it can cause the model to waste its capacity on reconstructing unimportant features, such as static backgrounds, while ignoring the visually small but important details in its learned representation (Anand et al., 2019; Kipf et al., 2019). For instance, in order to manipulate a small object in a complex scene, the model's ability to infer the object's shape carries more importance than inferring its color or reconstructing the static background.

To address this issue, we propose an approach which contrasts functions, rather than attempting to reconstruct them. The key idea is that two sets of examples of the same function should have similar latent representations, while the representations of different functions should be easily distinguishable. To this end, we propose a novel contrastive learning framework which learns by contrasting sets of input-output pairs (*partial observations*) of different functions. We show that this self-supervised training signal allows the model to meta-learn task-agnostic, low-dimensional representations of functions which are not only robust to noise but can also be reliably used for a variety of few-shot downstream prediction tasks defined on those functions. To evaluate the effectiveness of the proposed method, we conduct comprehensive experiments on diverse downstream problems including classification, regression, parameter identification, scene understanding and reinforcement learning.

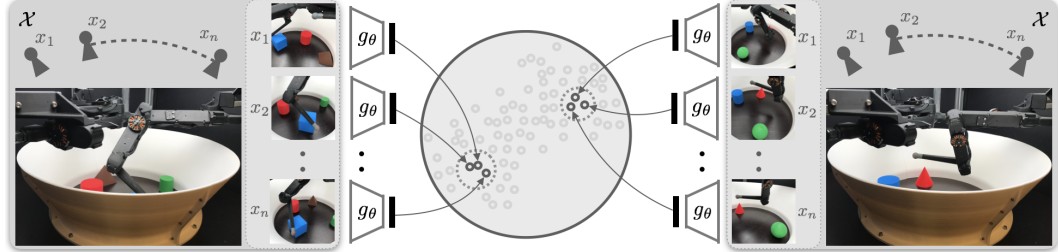

Figure 1: FCRL for self-supervised scene representation learning: The representations are learned by mapping the context points of each scene closer in the latent space while separating it from the other scenes. The context points correspond to the tuples of camera viewpoints $x_i \in \mathcal{X} := \mathbb{R}^d$ and the images taken from those viewpoints $y_i \in \mathcal{Y} \subseteq \mathbb{R}^{d'}$. Seen here, TriFinger (Wüthrich et al., 2020).

We consider different datasets, ranging from simple 1D and 2D regression functions to the challenging simulated and real-world scenes. In particular, we find that a downstream predictor trained with our (pre-trained) representations performs comparable or better than related methods on these tasks, including the ones wherein the predictor is trained jointly with the representation.

**Contributions.**

- Our key contribution is the insight that the good representation of an underlying data-generating function can be learned by aligning the representations of the samples that stem from it. This perspective of learning functions' representations is different from the typical regression methods and is new to our knowledge.

- We propose a novel contrastive learning framework, Function Contrastive Representation Learning (FCRL), to learn such representation of a function, using only its few observed samples.

- With experiments on diverse datasets, we show that the functions' representations learned by FCRL are not only robust to noise in inputs but also transfer well to multiple downstream problems.

## 2 PRELIMINARIES

### 2.1 PROBLEM SETTING

Consider a data-generating function $f : \mathcal{X} \to \mathcal{Y}$ with $\mathcal{X} = \mathbb{R}^d$ and $\mathcal{Y} \subseteq \mathbb{R}^{d'}$:

$$y = f(x) + \xi; \quad \xi \sim \mathcal{N}(0, \sigma^2) \tag{1}$$

where $\xi \sim \mathcal{N}(0, \sigma^2)$ is Gaussian noise. Let $O^f = \{(x_i, y_i)\}_{i=1}^N$ be a set of few observed samples of a function $f$, referred to as the context set and $\mathcal{T} = \{T_t\}_{t=1}^T$ be a set of unknown downstream tasks that can be defined on $f$. The downstream tasks $\mathcal{T}$ can take the form of classification, regression and/or reinforcement learning problems and the targets in each task $T_t$ vary accordingly. Our goal is to learn the representation of a given function $f$ from only the context set $O^f$, such that the representation retains the maximum useful information about $f$ and can interchangeably be used for multiple downstream tasks $\mathcal{T}$, defined on the same function (without requiring retraining).

**Few-shot Interpretation.** Note that the downstream tasks $\mathcal{T}$ are defined directly on the function's representation. Since the representation is inferred from a few observations $O^f$, the overall downstream problem becomes a few-shot prediction problem. In the following, we regard the downstream problems as few-shot problems, unless stated otherwise.

### 2.2 BACKGROUND

While there exist a variety of methods (Xu et al., 2019; Finn et al., 2017) that take different approaches towards few-shot learning for functions, a class of methods that is particularly relevant to our setting is that of conditional neural processes (CNPs) and NPs (Garnelo et al., 2018a;b).

**Conditional Neural Processes (CNPs).** The key proposal in CNPs (applied to few-shot learning) is to express a distribution over predictor functions given a context set. To this end, they first encode the context $O^f$ into individual representations $r_i = h_\Phi(x_i, y_i) \ \forall i \in [N]$, where $h_\Phi$ is a neural network. The representations are then aggregated via a mean-pooling operation into a fixed size

vector $r = 1/N(r_1 + r_2 + ... + r_N)$. The idea is that $r$ captures all the relevant information about $f$ from the context dataset $O^f$; accordingly, the predictive distribution is approximated by maximizing the conditional likelihood of the target distribution $p(y^T | x^T, O^f)$, where $T_t = \{(x_i^T, y_i^T)\}_{i=N}^{N+M}$ is the target set of size $M$.

**Generative Query Networks (GQN).** GQN (Eslami et al., 2018) can be seen as an extension of NPs for learning 3D scenes representations. The context dataset $O^f$ in GQN consists of tuples of camera viewpoints in 3D space ($\mathcal{X}$) and the images taken from those viewpoints ($\mathcal{Y}$). Like NPs, GQN learns to infer the latent representation of the scene (a function) by conditioning on the aggregated context and maximizing the likelihood of generating the target image corresponding to a target viewpoint.

### 2.3 MOTIVATION AND INTUITION

Learning about data-generating functions is ubiquitous in machine learning. Whether it is about learning scenes as deterministic functions that map camera viewpoints to the images (Eslami et al., 2018) or digital soil mapping where one is given a set of soil samples taken from some regions, and asked to predict the nature of soil in another region (Minasny & McBratney, 2016). In practice, we only observe a few views $O^f$ of such functions and the goal is to predict certain properties regarding those functions. More importantly, the observed samples often do not follow any order, are noisy and exhibit sampling biases, i.e. some samples are more informative than others. For instance, in learning a scene representation from multiple views, a single sampled view can have some objects occluded.

An ideal representation of a function, therefore, should not only be invariant to the permutations in the order of the observed samples, but also to noise and sampling of partial views $O^f$. Such an invariance to input order and sampling is important, as at test time we do not know the input order of the samples or how informative a certain sample is. In this work, we seek to learn such invariant representations of functions from only their few observations $O^f$. The inductive bias pertaining to permutation invariance is introduced via sum-decomposable encoder architecture (Zaheer et al., 2017). Whereas, invariance to the sampling of partial views is enforced via the contrastive objective. The idea is to align the representations of different samples stemming from the same function such that they are invariant to where they are sampled.

## 3 FUNCTION-CONTRASTIVE REPRESENTATION LEARNING (FCRL)

We take the perspective here that the sets of context points $O^f$ provide a partial observation of an underlying function $f$. Our goal is to find an encoder $g_{(\phi, \Phi)}$ which maps such partial observations to low-dimensional representations of the underlying function. The key idea is that a good encoder $g_{(\phi, \Phi)}$ should map different observations of the same function to be close in the latent space, such that they can easily be identified among observations of different functions. This motivates the contrastive-learning objective which we will detail in the following.

**Encoder Structure.** Since the inputs to the encoder $g_{(\phi, \Phi)}$ are sets, it needs to be permutation invariant with respect to input order and able to process inputs of varying sizes. We enforce this permutation invariance in $g_{(\phi, \Phi)}$ via sum-decomposition, proposed by (Zaheer et al., 2017). We first average-pool the point-wise transformations of $O^f$ to get the encoded representations.

$$r = \frac{1}{N} \sum_{i=1}^{N} h_\Phi(x_i, y_i) \quad \forall (x_i, y_i) \in O^f \tag{2}$$

where $h_\Phi(.)$ is the encoder network. We then obtain a nonlinear projection of this encoded representation $g_{(\phi, \Phi)}(O^f) = \rho_\phi(r)$. Note that the function $\rho_\phi$ can be any nonlinear function. We use an MLP with one hidden layer which also acts as the projection head for learning the representation. Similar to (Chen et al., 2020), we found that it is beneficial to define the contrastive objective on these projected representations $\rho_\phi(r)$ than directly on the encoded representations $r$. More details can be found in our ablation study Appendix A.1.

**Encoder Training.** At training time, we are provided with the few observations $O^{1:K}$ of $K$ functions. Each observation is a set of $N$ examples $O^k = \{(x_i^k, y_i^k)\}_{i=1}^N$. To ensure that different observations of the same functions are mapped to similar representations, we will now formulate a contrastive-learning objective. To apply contrastive learning, we need to create different observations of the same function $k$ by splitting each example set $O^k$ into $J$ subsets of size $N/J$. Defining $t_j := ((j-1)N/J, ..., jN/J)$

we hence obtain a split of $O^k$ into $J$ disjoint subsets of equal size

$$O^k = \cup_{j=1}^J O_{t_j}^k \text{ with } O_{t_i}^k \cap O_{t_j}^k = \emptyset \text{ if } i \neq j \tag{3}$$

where each subset $O_{t_j}^k$ is a partial observation of the underlying function $k$. We need at least two distinct observations ($J \geq 2$) per function for contrastive learning. Hence, $J$ is a hyper-parameter and can vary in the range of $[2, N]$ while ensuring the even division of $N$. Its value is empirically chosen based on the data domain. E.g. in 1D and 2D regression, the number of examples per observation is relatively large, since a single context point does not provide much information about the underlying function, whereas in scenes, a few (or even one) images provide enough information. We can now formulate the contrastive learning objective as follows:

$$L(\phi, \Phi, O^{1:K}) = \sum_{k=1}^K \sum_{1 \leq i < j \leq J} \left[ \log \frac{\exp\left(\text{sim}\left(g_{(\phi,\Phi)}(O_{t_j}^k), g_{(\phi,\Phi)}(O_{t_i}^k)\right)/\tau\right)}{\sum_{m=1}^K \exp\left(\text{sim}\left(g_{(\phi,\Phi)}(O_{t_j}^k), g_{(\phi,\Phi)}(O_{t_i}^m)\right)/\tau\right)} \right] \tag{4}$$

where $\text{sim}(a, b) := \frac{a^\top b}{\|a\|\|b\|}$ is the cosine similarity measure. Intuitively, it acts as a discriminatory function and predicts for each observation $O_{t_j}^k$ which of the $J$ observations it was jointly drawn with, by assigning high scores to the pairs jointly drawn and low to other pairs. In other words, the objective function in Equation (4) enables learning the differences between the distributions of $K$ underlying functions by discriminating between pairs of observations coming from the same function $k$, called *positives*, and observations stemming from different functions, called *negatives*. The $\tau$ is a temperature parameter which scales the scores, returned by the inner product. Similar to SimCLR (Chen et al., 2020), we find that temperature adjustment is important for learning good representations. We treat it as a hyperparameter and perform an ablation study in Appendix A.1. The overall FCRL algorithm is described in Algorithm 1.

Note that even when $J > 2$, we only consider two *positives* at a time for computing the contrastive loss i.e. for each view, we have only one positive match. We then compute the contrastive loss for all the pairwise combinations of the *positives* and then sum them up. For instance, when J=3, we have $^3C_2 = 3$ combinations of *positives*. The term $\sum_{1 \leq i < j \leq J}$ in Equation (4) refers to that summation over all the combinations.

### 3.1 Application to Downstream Tasks

Once representation learning using FCRL has been completed, $h_\Phi$ is fixed and can now be used for any few-shot downstream prediction task $\mathcal{T} = \{T_t\}_{t=1}^T$ that can be defined on the underlying data-generating function. To solve a particular downstream prediction problem, one only optimize a parametric decoder $p_\psi(.|r)$ which conditions on the learned representation $r$. Specifically, the decoder maps the representations learned in the previous step to the variable we are trying to predict in the given task. Depending on the nature of the downstream prediction task, the conditional distributions and the associated objectives can be defined in different ways.

## 4 Experiments

To illustrate the benefits of learning function representations without an explicit decoder, we consider four different experimental settings. In all the experiments, we first learn the encoder, and then fix it and optimize decoders for the specific downstream problems in the second stage.

**Baselines.** We compare the downstream predictive performance of FCRL based representations with the representations learned by the closest task-oriented, meta-learning baselines. For a fair comparison, all the baselines and FCRL have the same encoding architecture. For instance, for 1D and 2D regression functions, we considered CNPs and NPs as the baselines which have the same contextual embedding framework but optimize directly for the predictive distribution $p(y|x)$. Similarly for scene datasets, we took GQN as the baseline which explicitly learns to reconstruct scenes using a limited number of context sets.

### 4.1 1-D Sinusoidal Functions

In the first experiment, we consider a distribution over sinusoidal functions, proposed by (Finn et al., 2017). The amplitude and the phase of the functions are sampled uniformly from $[0.1, 0.5]$

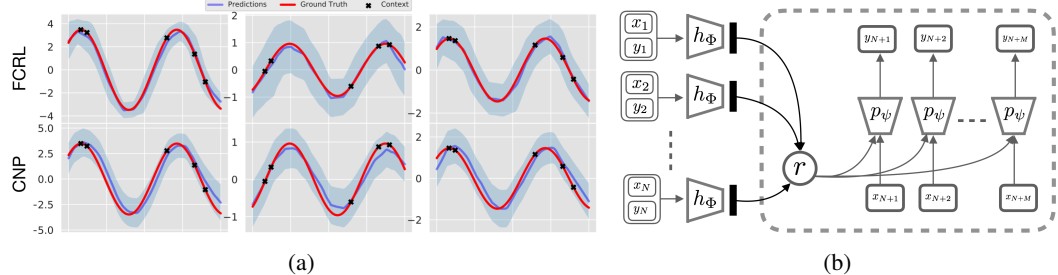

(a)                                    (b)

Figure 2: (a) Qualitative comparison of different methods on 5-shot regression task on three different sinusoid waves. The decoder trained on FCRL's representation is able to predict the correct form of the waves. (b) shows how we adapt the representation for the downstream few-shot regression task.

and $[0, \pi]$ respectively. To generate a context set of a particular function $f$, the $x$-coordinates are uniformly sampled from $[-5.0, 5.0]$ and then $f$ is applied to obtain the $y$-coordinates. The dataset contains $20,000$ training, $1000$ validation and $1000$ test functions. Once the representation is learned, we define two downstream tasks on the learned representation $\mathcal{T}_{1D} = \{T_{fsr}, T_{fspi}\}$, where $T_{fsr}$ corresponds to few-shot regression task and $T_{fspi}$ corresponds to few-shot parameter identification task. We use the same training data to train both the encoder in the representations learning phase and the decoders in the downstream transfer learning phase.

**Representation Learning Stage.** For training the encoder $g_{(\phi, \Phi)}$, we have a dataset $O = (O^k)_{k=1}^K$ at our disposition, where each $k$ corresponds to a function $f_k$ which has been sampled as described above. Each of these individual datasets $O^k = \{(x_i^k, y_i^k)\}_{i=1}^N$ consists of $N$ input-output pairs from that particular function $f_k$, such that $y_i^k = f_k(x_i^k)$. We fix the maximum number of context sets to 20 and the number of examples $N$ is chosen randomly in $[2, 20]$ for each $k$. The encoder $g_{(\phi, \Phi)}$ is then trained as described in Section 3 by splitting each context set $O^k$ into two observations i.e. $J = 2$.

### 4.1.1 DOWNSTREAM TASKS ON 1-D SINUSOIDAL FUNCTIONS

After training the encoder $g_{(\phi, \Phi)}$, we discard the projection head $\rho_\phi$ and use the trained encoder $h_\Phi$ to extract the representations. The decoders for the downstream tasks are trained as following.

**Few-shot Regression (FSR).** FSR for 1-D functions is a well-studied problem in meta-learning frameworks (Garnelo et al., 2018a; Finn et al., 2017; Kim et al., 2019; Xu et al., 2019). For each sampled function $f_k$, the target data is of the form $T_{fsr}^k = \{(x_i^T, y_i^T)\}_{i=1}^M$, where $M$ is the number of unseen target samples of function $k$. The number $M$ is chosen randomly in $[0, 20 - N]$. The goal is to predict the function values $f_k(x^T)$ at unseen inputs $x^T$, given some context $O^k$. The predictive distribution for $T_{fsr}^k$ is $P_\psi(y^T | x^T, O^k)$ and the decoder $p_\psi$ is trained to optimize the following conditional log probability

$$\min_\psi \mathbb{E}_{f_k \sim p(f)} \big[ \mathop{\mathbb{E}}_{N+M} [\log P_\psi(\{y^k\}_{i=1}^{N+M} | \{x^k\}_{i=1}^{N+M}, r^k)]\big] \tag{5}$$

where $r^k$ is the encoded representation of $f_k$, obtained by the context set $O^k$ and the trained encoder $h_\Phi$, following Equation (2). Figure 2(b) how the downstream decoder $p_\psi$ is trained on $r^k$. The decoder $p_\psi$ is a multi-layer perceptron with two hidden layers and is trained with the same training data as $h_\Phi$. The decoder learns to output the Gaussian mean and variance of $y^T$ to quantify uncertainty in the point estimates. The qualitative results on test sets in Figure 2(a) show that our model is able to quickly adapt with only 5 context points. We compare our method with CNP and NP and show that the predictions of our method are closer to the groundtruth, even though the encoder and decoder in both CNP and NP are explicitly trained to directly maximize the log likelihood to fit the wave.

**Few-shot Parameter Identification (FSPI).** The goal here is to identify the amplitude and phase of the sampled sine wave $f_k$. For each $f_k$, the target data is of the form $T_{fspi}^k = \{y_{amp}^k, y_{phase}^k\}$, where $y_{amp}^k$ and $y_{phase}^k$ are the amplitude and phase values of $f_k$. We train a linear decoder $p_\psi$ on top of the representations by maximizing the likelihood of the sine wave parameters. The predictive distribution is $P_\psi(T_{fspi}^k | O^k)$ and the objective is:

$$\min_\psi \mathbb{E}_{f_k \sim p(f)} [\log P_\psi(y_{amp}^k, y_{phase}^k | r^k)] \tag{6}$$

| Models | Few-shot Regression | | Few-shot Parameter Identification | |
|---|---|---|---|---|
| | **5-shot** | **20-shot** | **5-shot** | **20-shot** |
| NP | $0.310 \pm 0.05$ | $0.218 \pm 0.02$ | $0.0087 \pm 0.0007$ | $0.0037 \pm 0.0005$ |
| CNP | $0.265 \pm 0.03$ | $0.149 \pm 0.02$ | $0.0096 \pm 0.0007$ | $0.0049 \pm 0.0011$ |
| FCRL | $\mathbf{0.172 \pm 0.04}$ | $\mathbf{0.100 \pm 0.02}$ | $\mathbf{0.0078 \pm 0.0004}$ | $\mathbf{0.0032 \pm 0.0002}$ |

Table 1: Mean squared error for all the target points in 5 and 20 shot regression and parameter identification tasks on test sinusoid functions. The reported values are the mean and standard deviation of three independent runs.

where $r^k$ is the encoded representation of $f_k$, obtained by the context set $O^k$ and the trained encoder $h_\Phi$, following Equation (2). Similar to FSR, we use the same training data. In Table 1, we report the mean squared error for three independent runs, averaged across all the test tasks for 5-shots and 20-shots FSR and FSPI. In both prediction tasks, the decoders trained on FCRL representations outperform CNP and NP. More details on the experiment are given in Appendix G.

## 4.2 MODELING IMAGES AS 2-D FUNCTIONS

In the second experiment, we consider a harder regression dataset where images are modelled as 2-D functions. This form of image modeling has been considered in the context of image completion by (Garnelo et al., 2018a;b; Gordon et al., 2020; Kim et al., 2019). More specifically, the 2-D pixel coordinates of an image are considered as the inputs $x_i$ while the corresponding pixel intensities the outputs $y_i$. We consider images of MNIST hand-written digits (LeCun et al., 1998) where $x_i$ are the normalized pixel coordinates $[0,1]^2$ and $y_i$ the grayscale pixel intensities $[0,1]$. Similar to the previous section, we first describe how we learn the representations, and then we formulate two downstream prediction tasks on the learned representation: $\mathcal{T}_{2D} = \{T_{fsic}, T_{fscc}\}$, where $T_{fsci}$ corresponds to few-shot image completion and $T_{fscc}$ corresponds to few-shot content classification task. The training data consists of $60,000$ MNIST training samples whereas the validation is done with the $10,000$ test samples. We use the same data to train both the encoder in the representations learning phase and the decoders in the downstream transfer learning phase.

**Representation Learning Stage.** For training the encoder $g_{(\phi,\Phi)}$, we use the same training procedure as was used for 1-D functions. We have a context set $O = (O^k)_{k=1}^K$ of $K$ functions at our disposition, where each $k$ corresponds to an MNIST digit. Here, each individual context set $O^k = \{(x_i^k, y_i^k)\}_{i=1}^N$ corresponds to $N$ sampled pixels, where $x_i$ is the pixel location and $y_i$ is the corresponding pixel's intensity. We fix the maximum number of context sets to 200 and the number of examples $N$ is chosen randomly in $[2, 200]$ for each $k$. The encoder $g_{(\phi,\Phi)}$ is then trained as described in Section 3 by splitting each context set $O^k$ into 40 observations i.e. $J = 40$.

### 4.2.1 DOWNSTREAM TASKS ON 2-D FUNCTIONS

Again, after training the encoder $g_{(\phi,\Phi)}$, we discard the projection head $\rho_\phi$ and use the trained encoder $h_\Phi$ to extract the representations. The decoders for the downstream tasks are trained as following.

**Few-shot Image Completion (FSIC).** For few-shot image completion, we train a decoder $p_\psi$ to maximize the conditional likelihood of the target pixel values $T_{fsic}^k = \{(x_i^T, y_i^T)\}_{i=1}^M$, where $M$ is the number of unseen target pixels of function $f_k$. The number $M$ is chosen randomly in $[0, 200 - N]$. The conditional likelihood objective is the same as for FSR in 1D sinusoid functions i.e. Equation (5). The decoder is an MLP with two hidden layers as in CNP and NP. We use the same dataset to train it as was used to train the encoder.

Qualitative results of FSIC on test images are shown in Figure 3(a). It can be seen that the decoder trained on FCRL representations is able to predict the pixel intensities even when the number of context points is as low as 50, i.e., approximately $6\%$ of the image. We compare its performance against CNP which uses the same architecture. It is important to note that the feature extraction via FCRL is not explicitly carried out to optimize for this pixel-based reconstruction, and the decoder has to rely on the signal in the pre-trained representation for this complex image completion task. In other words, no gradient flows from the decoder to the encoder, and the former is trained independently of the latter. On the contrary, CNP jointly optimizes both encoder and decoder parameters for predicting

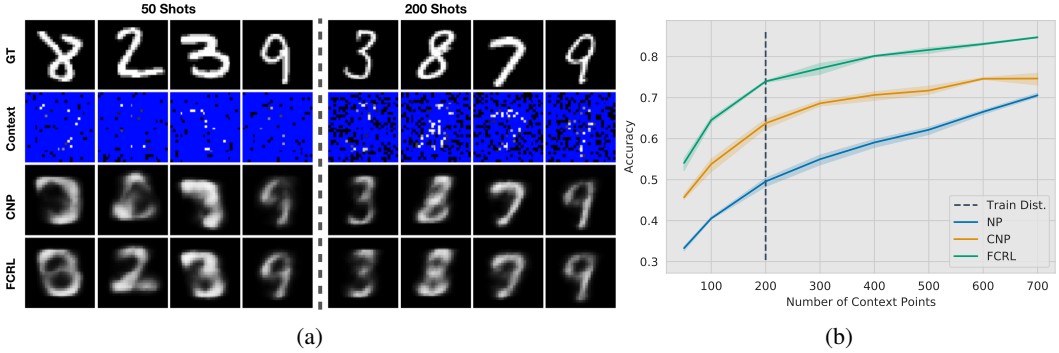

Figure 3: (a) Qualitative comparison of CNP and FCRL based few-shot image completion. The context is shown in the second row where target pixels are colored blue. FCRL appears to be slightly better at predicting the correct form of a digit in low-shot scenario of 50 context points only. (b) Quantitative evaluation of the models in terms of digit classification from the fixed number of context points (varying along the x-axis). FCRL achieves substantially higher accuracy.

the pixels values. Despite the fact, it appears that the FCRL-based reconstruction is slightly more accurate when using only 50 context points.

**Few-shot Content Classification (FSCC).** To test how much semantic knowledge each model captures in its representations, we devised a few-shot content classification task $T_{fscc}$. The goal is to predict the class of each MNIST digit, given its few randomly sampled pixel values as context $O^k$. The lack of spatial structure in the context points makes it a challenging problem. We consider the representations of all the functions trained in the previous section and train a linear decoder on top to maximize the likelihood of the corresponding class labels $T_{fscc} = \{y_{one\_hot}^k\}$ i.e.:

$$\min_{\psi} \mathbb{E}_{f_k \sim p(f)}[\log P_{\psi}(y_{one\_hot}^k | r^k)] \tag{7}$$

We use the same data as was used for training the encoders. Figure 3(b) shows the performance of three independently trained models for each method on MNIST validation set. FCRL has been able to recognize the semantic labels of the images much better than the competing methods for different number of fixed context points. This probably also explains why it is able to guess the correct form of digits in FSIC, albeit a little blurry.

### 4.3 Representing Scenes as Functions

Like (Eslami et al., 2018), we represent scenes as deterministic functions which map camera viewpoints to images. A specific scene (i.e. function) is given by the underlying factor of variations (e.g. the position of an object in the scene). To learn scene representations, we use two different datasets: the real-world robotics dataset, MPI3D (Gondal et al., 2019), and a robotics simulation dataset (*RLScenes*), which we created for the present work. Both datasets contain multiple views for each scene. In the MPI3D dataset, the multi-view setting is formulated by considering the images corresponding to the *camera height* factor as views, hence, there are three views per scene. RLScenes comprises of 36 views per scene, corresponding to 36 uniformly distributed 3D viewpoints. More details on the datasets are provided in Appendix E.

The purpose of using two different datasets for scenes is to test the generalization capability of the learned representations to two different types of downstream tasks. The set of downstream tasks on MPI3D correspond to scene understanding, $\mathcal{T}_{mpi3D} = \{T_v\}_{v=1}^6$, where $v$ corresponds to a factor of variation. On the other hand, RLScenes is used to test the generalization of the learned representations to a reinforcement learning downstream task.

**Scenes' Representation Learning Stage.** We use the same setting for learning the representations on both datasets. Each function $f_k$ corresponds to a scene here, and we randomly draw $N$ tuples of (viewpoints, views) as context sets $O^k = \{(x_i^k, y_i^k)\}_{i=1}^N$, where $x_i$ is the 3D camera viewpoints and $y_i$ is the corresponding image, taken from that viewpoints. We fix the maximum number of context sets to 3 in MPI3D dataset and 20 in RLScenes. The number of tuples drawn, $N$, is then chosen randomly in $[2, 3]$ for MPI3D and $[2, 20]$ for RLScenes. For more details, see Appendix F.

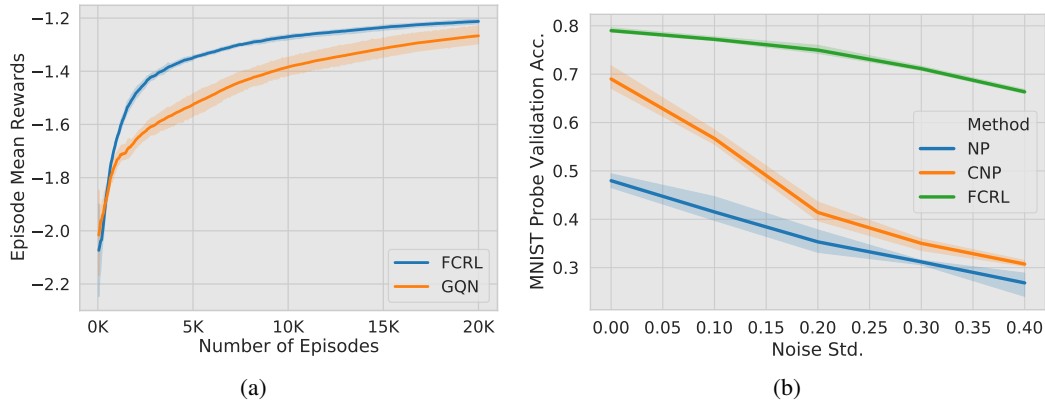

Figure 4: Quantitative Comparison of FCRL and GQN on MPI3D downstream classification tasks.

### 4.3.1 DOWNSTREAM TASKS ON SCENES

After training the encoder $g_{(\phi,\Phi)}$, we discard the projection head $\rho_\phi$ and use the trained encoder $h_\Phi$ to extract the representations $r^k$ and train decoders for downstream problems.

**Scene Understanding on MPI3D Dataset.** In MPI3D, each scene is described by 6 factors of variations. Therefore for each sampled scene $f_k$, we have 6 different tasks $\mathcal{T}_{mpi3D} = \{T_v^k\}_{v=1}^6$. Each task $T_v^k$ comprises of a discrete label for a factor of variation i.e. $T_v^k = \{y_v^k\}$. Hence, we train 6 linear decoders to maximize the likelihood of the labels corresponding to the factors, using the same objective as in Equation (7). Note that we use only one image to infer the representation $r^k$. Figure 4 shows the linear probes validation performance for five independently trained models on both GQN learned representations and FCRL learned representations. It can be seen that the representations learned by FCRL consistently outperform GQN for identifying all the factors of variations in scenes.

**Reinforcement Learning on RLScenes Dataset.** For RLScenes, instead of training linear decoders for downstream tasks, we train an RL agent to learn a control policy for the robot finger. The goal for the agent is to locate the object in the arena, reach it, and stay close to it for the remainder of the episode. We use the Soft-Actor-Critic (SAC) algorithm (Haarnoja et al., 2018) using the implementation from (Raffin et al., 2019) to learn a single MLP policy for all the joints of the robot, and predict their positions by having access only to the representation of the scene. These representations are inferred from a single view. The positions of the robot-finger and the object are reset at the beginning of every episode. It should be noted that unlike previous experiments, the downstream policy is not trained with the same dataset as was used to train the encoder. We again use the representations trained with GQN as our baselines and evaluate the performance on five independent runs. The results are shown in Figure 5(a). It can be seen that the RL agent, learned with FCRL representation clearly outperforms GQN-based RL agent in terms of learning a data-efficient policy, obtaining convergence level control performance with approximately 2 times fewer interactions with the environment.

Figure 5: (a) Comparison between GQN and FCRL on learning a data-efficient control policy for an object reaching downstream task. FCRL based representations clearly outperform GQN's representations. (b) Quantitative comparison for noise robustness on MNIST content classification downstream task. The representations learned with FCRL are much more robust to noise than GQN's.

### 4.4 ROBUSTNESS TO NOISE CORRUPTIONS

So far in our experiments, we have considered the functions to be noiseless. However, in real-world settings, the functions are often corrupted with noise which is manifested in their outputs. In this section, we investigate how robust different representations learning algorithms are to Gaussian noise with varying levels of noise corruption. We train all the models on the noisy data and evaluate the quality of the learned representation on downstream tasks, as defined above. We find that the representations learned by FCRL are far more robust to such noise corruptions than the competing methods Figure 5(b). In fact, the performance of CNPs, NPs and GQNs drop so much that downstream tasks achieve random accuracy. This might be because the representations are learned by optimizing to reconstruct the noisy outputs where signal to noise ratio is very low. On the other hand, FCRL learns by contrasting the set of examples, extracting the invariant features, and getting rid of any non-correlated noise in the input. See Appendix B, for noise robustness study on MPI3D dataset.

## 5 RELATED WORK

**Meta-Learning.** Supervised meta-learning can be broadly classified into two main categories. The first category considers the learning algorithm to be an optimizer and meta-learning is about optimizing that optimizer, for e.g., gradient-based methods (Ravi & Larochelle, 2016; Finn et al., 2017; Li et al., 2017; Lee et al., 2019) and metric-learning based methods (Vinyals et al., 2016; Snell et al., 2017; Sung et al., 2018; Allen et al., 2019; Qiao et al., 2019). The second category is the family of Neural Processes (NP) (Garnelo et al., 2018a;b; Kim et al., 2019; Eslami et al., 2018) which draw inspirations from Gaussian Processes (GPs). These methods use data-specific priors in order to adapt to a new task at test time while using only a simple encoder-decoder architecture. However, they approximate the distribution over tasks in terms of their predictive distributions which does not incentivize NP to fully utilize the information in the data-specific priors. Our method draws inspiration from this simple, yet elegant framework. However, our proposed method extracts the maximum information from the context which is shown to be useful for solving not just one task, but multiple downstream tasks.

**Self-Supervised Learning.** Self-supervised learning methods aim to learn the meaningful representations of the data by performing some pretext learning tasks (Zhang et al., 2017; Doersch et al., 2015). These methods have recently received huge attention (Tian et al., 2019; Hjelm et al., 2018; Bachman et al., 2019; Chen et al., 2020; He et al., 2019) mainly owing their success to noise contrastive learning objectives (Gutmann & Hyvärinen, 2010). At the same time, different explanations have recently come out to explain the success of such methods for e.g. from both empirical perspective (Tian et al., 2020; Tschannen et al., 2019) and theoretical perspective (Wang & Isola, 2020; Arora et al., 2019). The goal of these methods has mostly been to extract useful, low-dimensional representation of the data while using downstream performance as a proxy to evaluate the quality of the representation. In this work, we take inspiration from these methods and propose a self-supervised learning method which meta-learn the representation of the functions. Instead of using randomly augmented views of the data points, our self-supervised loss uses partially observed views, sampled from the underlying functions. More recently, (Doersch et al., 2020) pointed out the generalization issue in meta-learning algorithms, namely the supervision collapse, and proposed self-supervised pre-training to address this issue. Similar setup is used by (Medina et al., 2020) for prototypical networks (Snell et al., 2017). However, these methods study generalization to tasks defined on new datasets. In contrast to these methods, we study generalization in terms of downstream tasks, defined on functions.

## 6 CONCLUSION

In this work, we proposed a novel self-supervised representations learning algorithm for few-shot learning problems. We deviate from the commonly-used, task-specific training routines in meta-learning frameworks and propose to learn the representations of the relevant functions independently of the prediction task. Experiments on widely different datasets and the related set of downstream few-shot prediction tasks show the effectiveness of our method. The flexibility to reuse the same representation for different task distributions defined over functions brings us one step closer towards learning a generic meta-learning framework. In the future, we plan to adapt this framework to tackle multiple challenging few-shot problems such as object detection, segmentation, visual question answering etc. all of them using the same generic representations of the data-generating processes.

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

## A APPENDIX

### A.1 ABLATION STUDY

**Role of Number of Observations (J)**    The number of observations $J$ corresponds to the number of partial observations that we have of a functions $f^k$. Ideally, we only need two such observations to learn the representations via contrastive objective. However, it has been shown that having more positive pairs result in learning better representations (Chen et al., 2020; Tian et al., 2019). It also corresponds to reducing the amount of Bayesian surprise in the models, as more observations from the same function results in reducing the entropy.

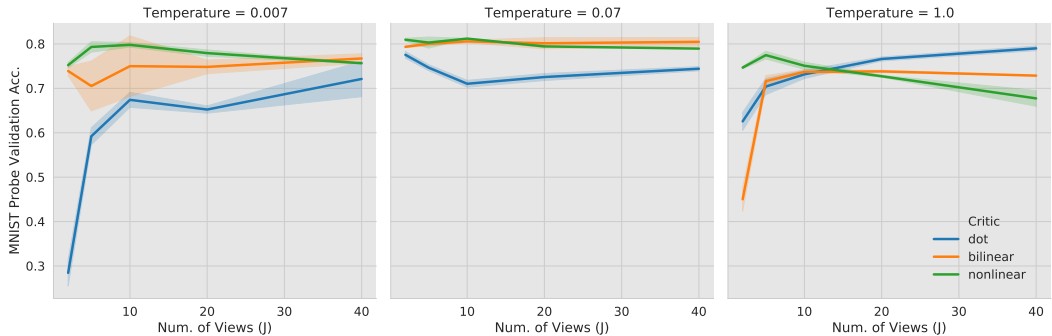

Figure 6: Ablation for observations ($J$), critics and temperature for learning FCRL on MNIST validation dataset.

It should be noted that in our setting, the number of observed context sets $N$ do not necessarily correspond to the number of observations $J$. This is because for some datasets, we aggregate the context points to get one partial observation. This is important for the simple 1D and 2D regression functions where a single context point does not provide much information, hence the individual partial observations need more than one context sets. In our setting, we treat $J$ as a hyperparameter whose optimal value varies depending on the function space. For instance, the MPI3D scene dataset has only three views per scene, therefore $J$ can not be greater than 3 and we keep it fixed. For 1D and 2D regression functions, we observe that for a fixed number of context points $N$, the optimal number of observations $J$ varies. For understanding the role of $J$ better in these experiments, we perform the ablation study on MNIST2D validation dataset where three models with different seeds are trained on the the varying number of $J$ while $N$ is fixed to 200, Figure 7.

It can be seen that varying the number of observations $J$ contributes to very small changes. However, its values seems to be highly dependent on the critic and the temperature parameter as well. For MNIST2D, we fixed $J = 10$. For training on RLScene dataset, we fixed the maximum number of context points to be $N = 20$ and found the optimal number of partial observations to be $J = 4$.

**Role of Critics and Temperature $\tau$.**    We regard the discriminative scoring functions, including the projection heads as critics. The simplest critic function does not contain any projection layer, regarded as *dot product* critic, where the contrastive objective is defined directly on the representations returned by the base encoder $h_\Phi$. However, recently the role of critics in learning better representations has been explored (Oord et al., 2018; Chen et al., 2020). Building on these findings, we evaluate the role played by different critics in learning the functions representations. Figure 7 shows the ablation for three different critics on MNIST2D validation dataset. It can be seen that the performance of critics is also linked with the temperature parameter $\tau$.

Such hyperparameter grid search (done for MNIST2D) is very expensive for the ablation studies on the bigger datasets i.e., the MPI3D and RLScenes datasets. We therefore performed a random sweep of 80 models with randomly selected hyper-parameter values for critic and temperature on MPI3D dataset. We did not find any pattern for the effect of temperature $\tau$ on the downstream tasks, however the pattern emerged for the class of critics. Figure 7 shows the ablation for critics on MPI3D

dataset. Again, it can be seen that the nonlinear critic performs good in extracting features which are useful for the downstream classification tasks. Because of this trend across two different datasets, we therefore performed all our experiments with nonlinear critic. The project head in nonlinear critics is defined as an MLP with one hidden layer and batch normalization in between.

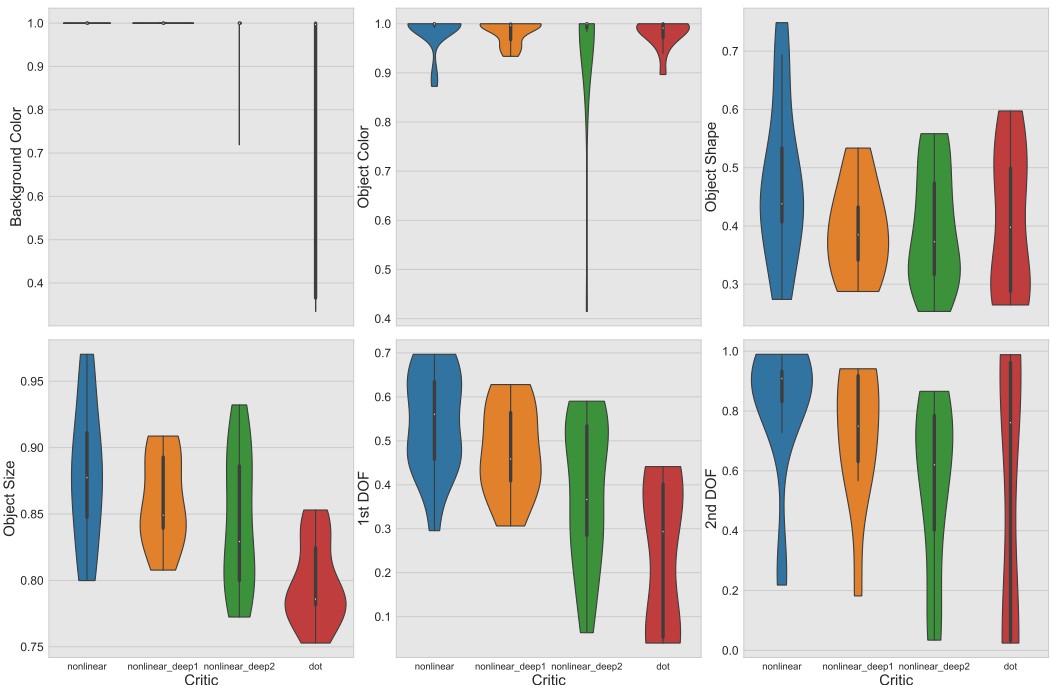

Figure 7: Ablation for critics on MPI3D validation dataset.

# B  ROBUSTNESS TO NOISE

Functions with additive noise have been well-studied, however, the contemporary literature in meta-learning mostly considers them to be noise free. In this work, we explore whether the learned representations of the functions are prone to the additive Gaussian noise. We consider the form of the function as given in Equation (1) and vary the standard deviation of the added noise. It can be seen that with the increased level of noise the features in the image start to diminish, shown in the Figure 8. GQN approach the learning problem by reconstructing these noisy images where the signal is already very weak. On the other hand, FCRL learns to contrast the scene representations with other scenes without requiring any reconstruction in the pixel space. This helps it in extracting invariant features in the views of a scene, thus getting rid of any non-correlated noise in the input.
In addition to the analysis on MNIST 2D regression task in Figure 5, we test the performance of these representations learning algorithm on MPI3D factors identification task in Figure 9. It can be seen that FCRL representations can recover the information about the data factors, even in the extreme case where the noise level is very high (standard deviation of 0.2). Whereas, GQN performs very poorly such that the downstream probes achieve the random accuracy.

# C  PSEUDOCODE FOR SCRL ALGORITHM

For the sake of completeness, we provide the pseudocode for FCRL representations learning algorithm:

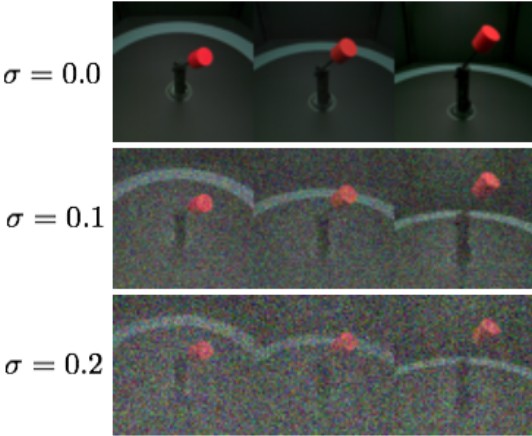

Figure 8: MPI3D dataset with the varying level of additive noise.

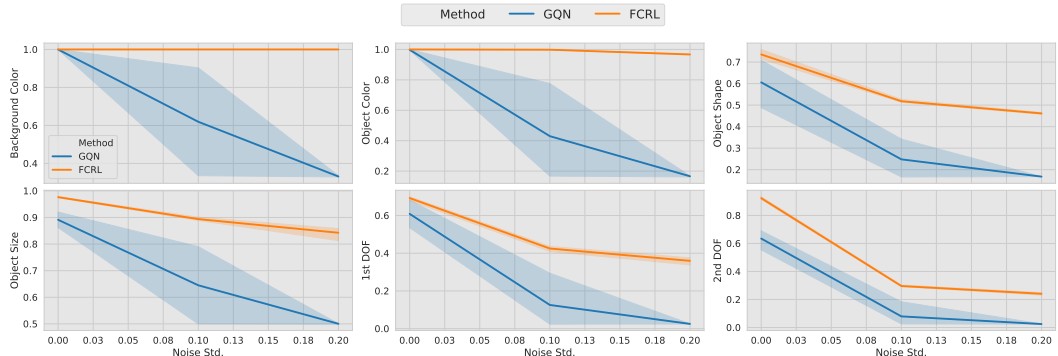

Figure 9: Quantitative comparison for noise robustness on MPI3D downstream tasks.

---

**Algorithm 1:** FCRL algorithm.

---

**input:** batch size $K$, a dataset of functions with input-output pairs $O^f = (\boldsymbol{x}, \boldsymbol{y})$, maximum number of context points $M$, constant $\tau, J$, networks $h_\Phi, \rho_\phi$.

**for** sampled minibatch of $K$ functions $\{O^f\}_{k=1}^K$ **do**

  draw a number N from range [2, M]

  # define observation size

  $t_j := ((j-1)N/J, ..., jN/J) \,\forall j = 1, \ldots, J$

  **for all** $k \in \{1, \ldots, K\}$ **do**

    draw $N$ context points $O^k = \{(x_i, y_i)\}_{i=1}^N$

    # split context into disjoint observations

    $O^k = \cup_{j=1}^J O_{t_j}^k$ with $O_{t_i}^k \cap O_{t_j}^k = \emptyset$ if $i \neq j$

    **for** $i \in \{1, \ldots, J\}$ and $j \in \{1, \ldots, J\}$**do**

      $z_i^k = \rho_\phi\big(\frac{1}{|O_{t_i}|} \sum_{m=1}^{|O_{t_i}|} h_\Phi(O_{t_i m}^k)\big)$

      $z_j^k = \rho_\phi\big(\frac{1}{|O_{t_j}|} \sum_{m=1}^{|O_{t_j}|} h_\Phi(O_{t_j m}^k)\big)$

      $s_{i,j}^k = \boldsymbol{z}_i^\top \boldsymbol{z}_j / (\|\boldsymbol{z}_i\| \|\boldsymbol{z}_j\|) \,\forall\, i \neq j$

    **end for**

  **end for**

  $\mathcal{L}(\Phi, \phi, O^{1:K}) = \sum_{k=1}^K \sum_{1 \leq i < j \leq J} - \log \frac{\exp(s_{i,j}^k/\tau)}{\sum_{m=1}^K \exp(s_{i,m}^k/\tau)}$

  update networks $h_\Phi$ and $\rho_\phi$ to minimize $\mathcal{L}$.

**end for**

**return** base encoder network $h_\Phi(\cdot)$

---

# D    ESTIMATING DENSITY RATIOS CORRESPONDING TO THE FUNCTIONS

The contrastive objective in Equation (4), in essence, tries to solve a classification problem i.e. to identify whether the given observation $O^i$ comes from the function $f^i$ or not. The supervision signal is provided by taking another observation $\hat{O}$ from the same function $f^i$ as an anchor (a target label), thus making it a self-supervised method. This self-supervised, view-classification task, for a function $f^i$, leads to the estimation of density ratios between the joint distribution of observations $p(O^1, O^2|i)$ and their product of marginals $p(O^1|i)p(O^2|i)$. This joint distribution in turn corresponds to the joint distribution of the input-output pairs of the function $p(x, y|i)$. This way of learning a function's distribution is different from the typical regression objectives, which learn about a given function $f^i$ by trying to approximate the predictive distribution $p(y|x)$.

By assuming the universal function approximation capability of $g_{(\phi, \Phi)}$, and the availability of infinitely many functions $f^k \sim p(f)$ with fixed number of context points $N$ each, the model posterior learned by the optimal classifier corresponding to Equation (4) would be equal to the true posterior given by Bayes rule.

$$f^k \sim P(f) \quad \forall \ k \in \{1, .., K\} \tag{8}$$

$$O^k \sim P(O|f^k) \quad \forall k \in \{1, .., K\} \tag{9}$$

$$i \sim \mathcal{U}(K) \tag{10}$$

$$\hat{f} = f^i \tag{11}$$

$$\hat{O} \sim P(O|\hat{f}) \tag{12}$$

$$p(i|O^{1:K}, \hat{O}) = \frac{p(O^{1:K}, \hat{O}|i)p(i)}{\sum_i p(O^{1:K}, \hat{O}|i)p(i)} \tag{13}$$

$$= \frac{p(O^i, \hat{O}|i)p(i) \prod_{k \neq i} p(O^k|i)p(\hat{O}|i)}{\sum_j p(O^j, \hat{O}|j)p(j) \prod_{k \neq j} p(O^k|j)p(\hat{O}|j)} \tag{14}$$

$$= \frac{\frac{p(O^i, \hat{O}|i)}{p(O_i)p(\hat{O})}p(i)}{\sum_j \frac{p(O^j, \hat{O}|j)}{p(O_j)p(\hat{O})}p(j)} \tag{15}$$

The posterior probability for a function $f^i$ is proportional to the class-conditional probability density function $p(O^i, \hat{O}|i)$, which shows the probability of observing the pair $(O^i, \hat{O})$ from function $f^i$. The optimal classifier would then be proportional to the density ratio given below

$$\exp(sim_{(\phi, \Phi)}(\hat{O}, O^i)) \propto \frac{p(O^i, \hat{O})}{p(O_i)p(\hat{O})} \tag{16}$$

Similar analysis has been shown by the (Oord et al., 2018) for showing the mutual information perspective associated with self-supervised contrastive objective (infoNCE). The joint distribution over the pair of observations correspond to the distribution of the underlying function $f^i$. Thus, given some observation of a function, an optimal classifier would attempt at estimating the true density of the underlying function.

# E    SCENES' DATASETS

**MPI3D Dataset.**    The MPI3D dataset (Gondal et al., 2019) is introduced to study transfer in unsupervised representations learning algorithms. The dataset comes in three different formats, varying in the levels of realism i.e. real-world, simulated-realistic and simulated-toy. Each dataset contains $1,036,800$ images of a robotic manipulator each, encompassing seven different factors of variations i.e., object colors (6 values), object shapes (6 values), object sizes (2 values), camera heights (3 values), background colors (3 values), rotation along first degree of freedom ((40 values)) and second degree of freedom ((40 values)). Thus, each image represents a unique combination of all the factors. See Figure 10(a) to see a sample of the dataset.

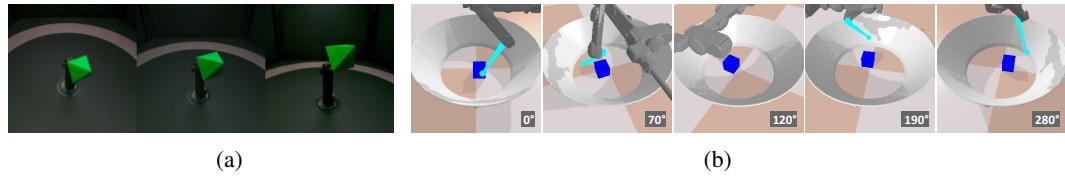

(a)                                                 (b)

Figure 10: Datasets for Scenes Representation Learning (a) MPI3D (Gondal et al., 2019) has three camera viewpoints, with images of a robotics arm manipulating an object. (b) RLScenes has 36 possible camera viewpoints for capturing an arena consisting of a robot finger and an object.

In this work, we consider the real-world version of the dataset. The multi-view setting is formulated by considering the images of a scene captured by three different cameras, placed at different heights. This effectively gives us $345,600$ scenes with three views each. We split the dataset into training and validation chunks, where the training dataset contains $310,000$ scenes and the validation dataset contains the rest $35,600$ scenes, approximately $10\%$ of the dataset.

**RLScenes.** The RLScenes dataset is generated in simulation using (Joshi et al., 2020) for a single 3-DOF robotic manipulator in a 3D environment. The dataset consists of $40,288$ scenes, each scene parametrized by: object colors (one of $4$), robot tip colours (one of $3$), robot positions (uniformly sampled from the range of feasible joint values), and object positions (uniformly sampled within an arena bounded by a high boundary as seen in Figure 10). Each scene consists of 36 views, corresponding to the uniformly distributed camera viewpoints along a ring of fixed radius and fixed height, defined above the environment. As can be seen in Figure 10, the robot finger might not be visible completely in all the views, or the object might be occluded in some view. The 36 views help by capturing a $360\deg$ holistic perspective of the environment. First a configuration of the above scene parameters is selected and displayed in the scene, then the camera is revolved along the ring to capture its multiple views. For learning the scene representations via both FCRL and GQN, we split the dataset into 35000 training and 5288 validation points.

## F    DETAILS OF EXPERIMENTS ON SCENE REPRESENTATION LEARNING

For learning the scene representations for both MPI3D dataset and RL Scenes, we used similar base encoder architecture. More specifically, we adapted the "pool" architecture provided in GQN (Eslami et al., 2018), as it has been regarded to exhibit better view-invariance, factorization and compositional characteristics as per the original paper. We further augmented this architecture with batch-normalization. The architecture we use is as in Figure 11:

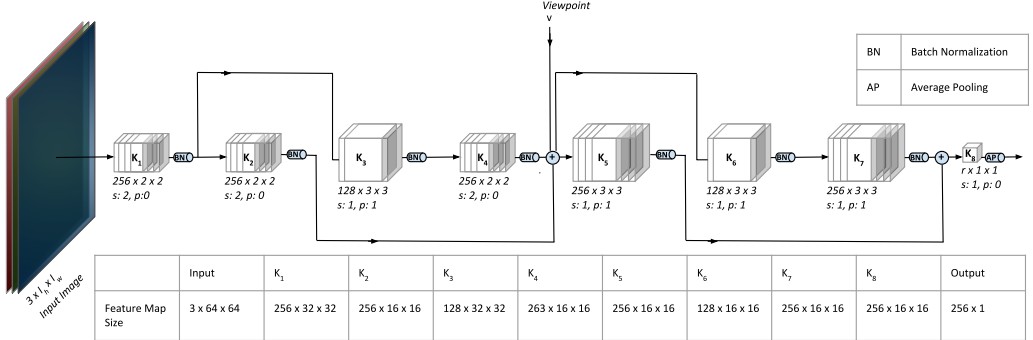

Figure 11: Network Architecture used for learning representations for the scenes' datasets.

### F.1    LEARNING SCENES FOR MPI3D DATASET.

Since we have three view per each scene in MPI3D dataset, therefore we are restricted in defining the number of context points and the number of views. In all our experiments the number of context points $N$ is fixed to three while the number of views $J$ is set to two. In contrast to the experiments

for regression datasets where more datapoints per each view resulted in better representations, the restriction to a single image view in MPI3D dataset did not hurt the representation quality, measured in terms of downstream performance. In the downstream experiments on MPI3D, we use only one image to train and validate the probes. Due to the limitation of available views, we could not measure the effect of varying the number of views on the downstream performance.

Even though no explicit structural constraint was imposed for learning the representation. The FCRL algorithm implicitly figures out the commonality between the factors in different scenes. We visualize these latent clusterings in the Figure 12. We plot the 2D TSNE embeddings of the 128D representations inferred by the model. Thereafter, to visualize the structure corresponding to each factor, we only vary one factor and fix the rest of them except for first degree of freedom and second degree of freedom factors. A clear structure can be seen in the learned representations.

**Implementation Details.** We use the GQNs 'pool' architecture with batch normalization as encoder. As mentioned in the ablation study, we did a random sweep over the range of hyperparameters and selected the best performing model. Further details on the hyperparameters is provided in Table 2.

## F.2 LEARNING SCENES FOR RLSCENES DATASET.

To train the FCRL encoder, we randomly sample the number of views from each scene to lie within the range [2, 20]: upper-bounded by 20 to restrict the maximum number of images per scene to be the same as that used in (Eslami et al., 2018), and lower bounded by 2 in case just the one view is not from a suitable angle. So, here, the maximum number of context points $N$ is 20. The number of subsets $J$ is set to 8. In the downstream reinforcement learning task, we use only one image to train the policy network, as is the usual practice, and the same as (Eslami et al., 2018). We kept the joint ranges from which joint positions are uniformly sampled to randomly reset the robot at the beginning of every episode while training the policy network to be the same as the ranges used for sampling the robot position while generating the dataset to train the FCRL encoder. These joint ranges are selected so as to ensure that there are more scenes in which the robot finger is visible. However, in order to not constrain the agent's exploration, we let the action space for training the reaching agent to be less constrained, and be able to explore the entire range from $-pi$ to $pi$. So, effectively, the space seen by the robot during the training of the representations is a subspace of that seen while inferring the representations from the environment used for this downstream reaching task. Interestingly, the inferred representations can also work effectively on unseen robot configurations as demonstrated by the success of the reacher.

**Implementation Details.** Similar to the encoder training for MPI3D scenes, we learned the encoder for RLScenes. However, since the downstream tasks is a reinforcement learning task, it was hard to judge the quality of representations. Therefore, we took some insights from the MPI3D experiments and selected the model, trained with hyperparameters, which performed the best on the RL downstream tasks. Further details on the hyperparameters is provided in Table 2.

## G DETAILS OF EXPERIMENTS ON 1D FUNCTIONS

**Implementation Details.** We used the same encoder architecture for our method and the baselines (Garnelo et al., 2018a;b) in all experiments. For 1D and 2D functions, the data is fed in the form of input-output pairs $(x, y)$, where $x$ and $y$ are 1D values. We use MLPs with two hidden layer to encode the representations of these inputs. The number of hidden units in each layer is $d = 50$. All MLPs have relu non-linearities except the final layer, which has no non-linearity.
*Encoder*: Input(2) $\rightarrow$ 2 $\times$ (FC(50), ReLU) $\rightarrow$ FC(50).
While learning the representations of sinusoid functions with FCRL, we also scale the output scores with temperature to be 0.07. We used the following hyper-parameter settings to train an encoder with FCRL.

**Downstream Tasks.** To learn the subsequent task-specific decoders on the representations, we adapted the same data processing pipeline as above. For 1D functions, we train decoders for two different tasks: *few-shot regression* and *few-shot parameter identification*. The decoders for each task

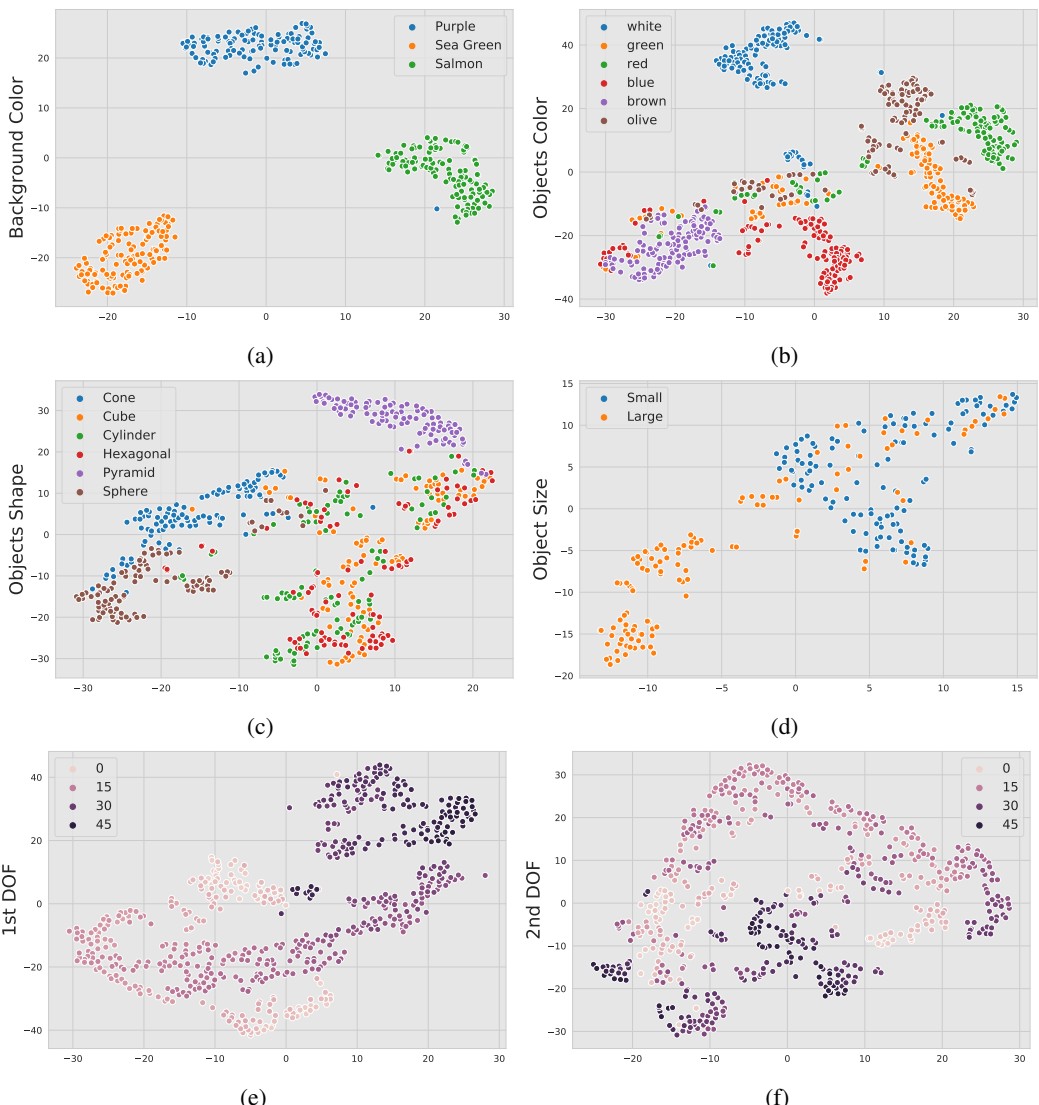

Figure 12: 2D-TSNE projections of 128 dimensional representations learned by FCRL on the MPI3D dataset. Each individual plot exhibits the latent structure corresponding to the factor mentioned above. The plot for each factor is generated by varying the factor of interest and keeping the rest of the factors fixed, except the factors of the first degree of freedom and the second degree of freedom.

Table 2: Hyperparameters Settings for Scene Representation Learning Experiments.

| Parameter | Values | Parameter | Values |
|---|---|---|---|
| Batch size | 64 | Batch size | 32 |
| Representation dimension | 128 | Representation dimension | 256 |
| Temperature: $\tau$ | 0.88 | Temperature: $\tau$ | 0.46 |
| Number of subsets: $J$ | 2 | Number of subsets: $J$ | 8 |
| Max number of context points: $N$ | 3 | Max number of context points: $N$ | 20 |
| Epochs | 100 | Epochs | 100 |
| Critic | Nonlinear | Critic | Nonlinear |
| Objective | NCE | Objective | NCE |
| Optimizer | Adam | Optimizer | Adam |
| Adam: beta1 | 0.9 | Adam: beta1 | 0.9 |
| Adam: beta2 | 0.999 | Adam: beta2 | 0.999 |
| Adam: epsilon | 1e-8 | Adam: epsilon | 1e-8 |
| Adam: learning rate | 0.0005 | Adam: learning rate | 0.0005 |
| Learning Rate Scheduler | Cosine | Learning Rate Scheduler | Cosine |
| Number of workers | 10 | Number of workers | 10 |
| Batch normalization | Yes | Batch normalization | Yes |

(a) Hyperparameters to train FCRL based encoder on the MPI3D Dataset.

(b) Hyperparameters to train FCRL based encoder on the RLScenes Dataset.

Table 3: Hyperparameters Settings for Sinusoid Experiments.

| Parameter | Values | Parameter | Values |
|---|---|---|---|
| Batch size | 256 | Batch size | 256 |
| Latent space dimension | 50 | Epochs | 30 |
| Temperature: $\tau$ | 0.07 | Critic | Nonlinear |
| Number of subsets: $J$ | 2 | Optimizer | Adam |
| Max number of context points: $N$ | 20 | Adam: beta1 | 0.9 |
| Epochs | 30 | Adam: beta2 | 0.999 |
| Critic | Nonlinear | Adam: epsilon | 1e-8 |
| Optimizer | Adam | Adam: learning rate | 0.001 |
| Adam: beta1 | 0.9 | Learning Rate Scheduler | Cosine |
| Adam: beta2 | 0.999 | | |
| Adam: epsilon | 1e-8 | | |
| Adam: learning rate | 0.0003 | | |
| Learning Rate Scheduler | Cosine | | |

(b) Hyperparameters to train FSR Decoder on FCRL learned representations.

(a) Hyperparameters to train FCRL based encoder for 1D sinusoid functions.

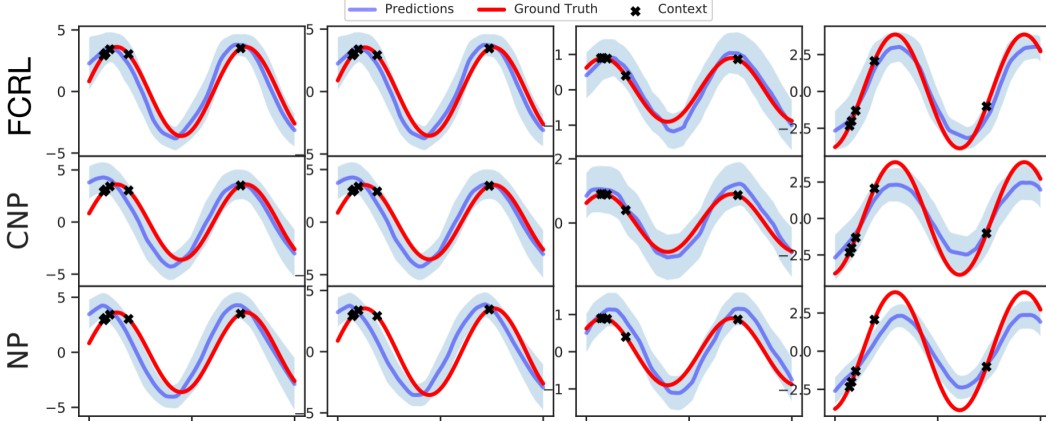

Figure 13: Additional results on 5-shot sinusoid regression. Each column corresponds to a different sinusoid function where only 5 context points are given. The predictions of the decoder trained on FCRL based encoder are closer to the groundtruth.

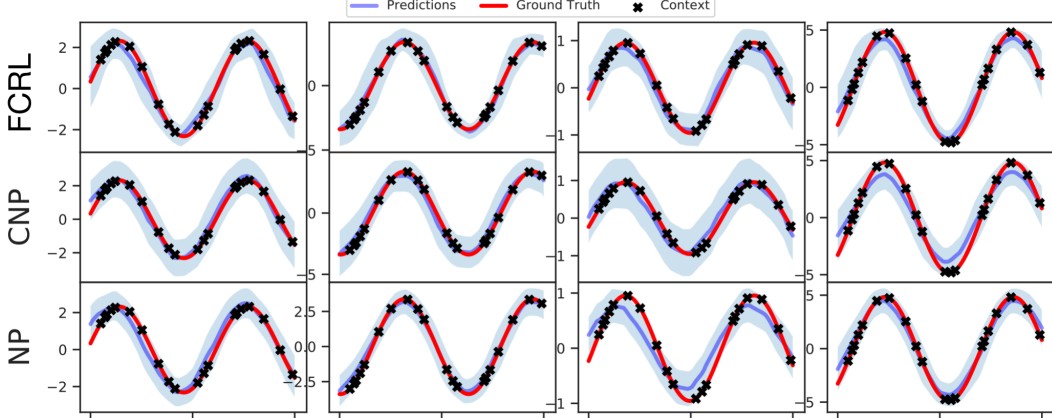

Figure 14: Additional results on 20-shot sinusoid regression. Each column corresponds to a different sinusoid function where only 20 context points are given. The predictions of the decoder trained on FCRL based encoder are comparable to CNP and better than NP.

are trained with the same training dataset as was used to train the encoders. The training procedure for both downstream tasks on sinusoid functions is as follows

- For Few-Shot Regression (FSR), we use an MLP architecture with two hidden layers. The same architecture are used in CNP (Garnelo et al., 2018a), however in CNP the decoder and encoder are trained jointly. All the baselines and our model are trained for the same number of iterations. We used slightly higher learning rate to train the decoder as the training converges quite easily.
  *FSR Decoder*: Input(50) $\to 2 \times$ (FC(50), ReLU) $\to$ FC(1) .

- For Few-Shot Parameter Identification (FSPI), we train a linear decoder without any activation layers on the representations learned via FCRL and the baseline methods. The decoder is trained for only one epoch.
  *FSPI Decoder*: Input(50) $\to$ FC(1) .

**Additional Results.** In Figure 13 and Figure 14, we provide additional results on 5-shot regression on test sets and compare the results with CNP and NP. The curves generated by the decoder using FCRL learned representations are closer to the groundtruth. The difference is evident in 5-shot experiments which supports the quantitative results in Table 1.

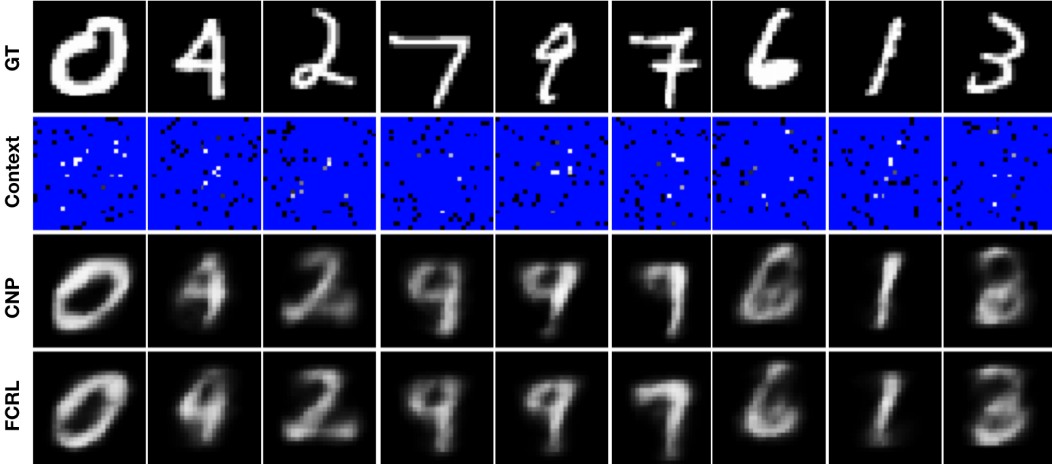

Figure 15: Additional results on 50-shot mnist image completion. The context is shown in the second row where target pixels are colored blue. Predictions made by a decoder trained on FCRL based encoder are slightly better than the CNP in terms of guessing the correct form of digits.

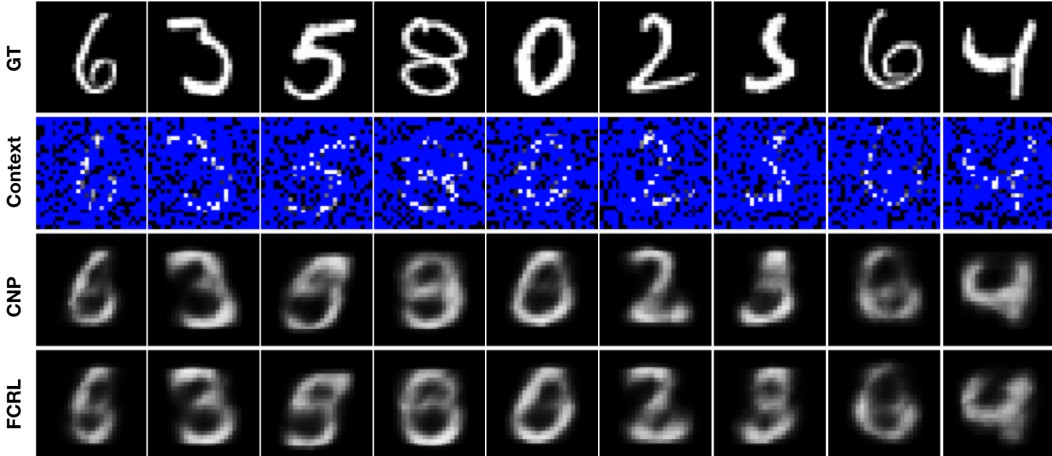

Figure 16: Additional results on 200-shot mnist image completion. The context is shown in the second row where target pixels are colored blue. Predictions made by a decoder trained on FCRL based encoder are comparable to CNP.

## H    DETAILS OF EXPERIMENTS ON 2D FUNCTIONS

**Implementation Details.**    In this experiment, we treat MNIST images as 2D functions. We adapt the architectures of encoders and decoders from the previous 1D experiments. However, due to the increased complexity of the function distributions we increase the number of hidden units of MLP to $d = 128$. Moreover, the input $x$ is 2D as it corresponds to the cartesian coordinates of an image. The hyperparameter settings to train FCRL based encoder on such 2D function is given in Table 4.

### H.1    DOWNSTREAM TASKS.

We consider two different downstream tasks for the representations learned on 2D functions: *few-shot image completion* and *few-shot content classification*. A separate decoder is trained for both of these tasks.

- For Few-Shot Image Completion (FSIC), we use an MLP based decoder with two hidden layers. The decoder is trained on the same training data for the same number of iterations. Details are given in Table 4(b).

Table 4: Hyperparameters Settings for MNIST as 2D Functions Experiment.

| Parameter | Values |
|---|---|
| Batch size | 16 |
| Latent space dimension | 128 |
| Temperature: $\tau$ | 0.007 |
| Number of subsets: $J$ | 40 |
| Max number of context points: $N$ | 200 |
| Epochs | 100 |
| Critic | Nonlinear |
| Optimizer | Adam |
| Adam: beta1 | 0.9 |
| Adam: beta2 | 0.999 |
| Adam: epsilon | 1e-8 |
| Adam: learning rate | 0.0006 |
| Learning Rate Scheduler | Cosine |

(a) Hyperparameters to train FCRL based encoder for 2D functions.

| Parameter | Values |
|---|---|
| Batch size | 16 |
| Epochs | 100 |
| Critic | Nonlinear |
| Optimizer | Adam |
| Adam: beta1 | 0.9 |
| Adam: beta2 | 0.999 |
| Adam: epsilon | 1e-8 |
| Adam: learning rate | 0.001 |
| Learning Rate Scheduler | Cosine |

(b) Hyperparameters to train Few-Shot Image Completion (FSIC) Decoder trained on FCRL learned representations.

- For Few-Shot Content Classification (FSCC), we train a linear regression on top of the representations obtained by both FCRL and the baselines. The decoder is trained for only one epoch.

**Additional Results.** In Figure 15 and Figure 16, we provide additional results for 50-shot and 200-shot image completion. We can see that the the results from FCRL based decoder consistently perform better than CNP in low-shot scenario of 50 context points. In 200-shot scenario, the results look comparable to CNP.

## I   SOFTWARE PROGRAMS USED

The following software programs were used for the work done in this paper: PyTorch (Paszke et al., 2019), Weights and Biases (Biewald, 2020), PyBullet (Coumans & Bai, 2016–2019), Stable Baselines (Raffin et al., 2019), Gym (Brockman et al., 2016), Jupyter Notebook (Kluyver et al., 2016), Matplotlib (Hunter, 2007), Seaborn (Waskom et al., 2017), Scikit-learn (Pedregosa et al., 2011), NumPy (Harris et al., 2020), Python3 (Van Rossum & Drake, 2009).

