# OpenReview forum: "Function Contrastive Learning of Transferable Representations"
_ICLR.cc/2021/Conference — Reject_

### Official Review · AnonReviewer3 · 2020-10-27
**I am completely lost. (update)**

**Rating:** 5
**Confidence:** 3

**Review:**

As far as I can tell, this paper proposes to use contrastive learning to solve few-shot learning -- not just few-shot learning, but meta "task-wise prediction" as well (predict some properties of the task itself). The method learns representation of each task, by forcing random instantiations of the same tasks to have similar representations. The method tests on sine-function fitting + param prediction, image (MNIST) inpainting + classification, and (robot) scene understanding + RL (both are meta "task-wise prediction").

Pro:
- Method seems reasonable and improves performance over compared methods (NP and CNP).
- Highlights an interesting point of view (which I think is relatively novel -- it's glossed over in the NP paper) that treats location-value prediction for pixels in different images as different tasks to test meta-learning algorithms. This change of view encourages the analogy and idea of using e.g. self-supervised learning or inpainting methods for meta-learning.

Con:
- The goal and motivation of this paper is lost to me. The abstract and intro does not inform readers of the goal of the paper and how it can be used in applications (such as ImageNet meta-learning). It seems to be all of (1) few-shot learning, (2) extension of existing works NP & CNP, and (3) learning representations for down-stream tasks. But it seems to be suboptimal for all of these goals.
  (1) If this is few-shot learning, then it would make sense to compare to other few-shot methods, e.g. ones using meta-learning.
  (2) If this is an extension of NP and CNP, then this method has no grounding probabilistically. It cannot provide estimates such as P(x|y), for example, which is the point of these methods. It is not very interesting that non-generative methods can outperform NP/CNP which are generative models.
  (3) If this is learning representations for down-stream tasks, the the down-stream-focus of the experiment setup violates the "few-shot" part since only the original formulation (e.g. image inpainting) is few-shot. For down-stream tasks (e.g. image classification) this is rather a "sparse input" problem. Despite the method being about learning representations, it does not compare to regular methods that treat images as images, not a set of (location,value) tuples. For example, a regular CNN may do better than the results reported in Figure 5.
  (4) If this is a method that can do both few-shot and learning representations, it should be still compared to regular few-shot and representation learning methods, especially because there are so many papers that do few-shot and representation learning, independently or combined.
- The motivation is quite similar to self-supervised learning, when considering down-stream applications where we treat each "task" as samples again.
- The statistical significance is confusing. For example, Table 1 has 0.0087±0.0007 and 0.0078±0.0004 being significantly different with 3 runs, which does not sound plausible.

Other minor issues:
- The method assumes each task is different. If there are many duplicated tasks (e.g. very similar images) then making each task representation different from all others would be suboptimal.
- In eq. (5), it seems the input for downstream tasks include the pixels/view-images, so it will be unclear if the knowledge is from X or $h_\Phi(D^T)$.

Summary:
Although viewing location-value prediction for pixels in different images as different tasks for meta-learning is really interesting, the paper fails to motivate its goal, and by extension, designing convincing experimental comparisons. I will be reserving judgement and lean towards rejection.

Update post-rebuttal:
The rebuttal clarified the motivation, but has yet to address the flaws that was associated with the choice of motivation and positioning among related work. Unfortunately that means I will not be changing my score.

---

> ### Author Response · Authors · 2020-11-17
> **First response to R3 (part 1)**
>
> Hi Reviewer 3, thank you for reviewing our work. We are currently working on the manuscript to reflect your suggestions; meanwhile, please find a detailed first response to your review below.
>
> > The goal and motivation of this paper is lost to me. The abstract and intro does not inform readers of the goal of the paper and how it can be used in applications (such as ImageNet meta-learning).
>
> Our goal is to learn the representation of a given function from a **few** observed pairs $O^f =\\{(x_i, y_i)\\}_{i=1}^N$ only once such that this representation can be used interchangeably for multiple downstream problems defined on the same function (without requiring retraining).
>
> In this paper, we consider a distribution over functions, in that we learn to encode individual functions to their latent representations. In the case of ImageNet, this is less straight forward, but still conceivable. Consider for instance a (hypothetical) subset of ImageNet with pictures of cats sitting on sofas. Now, we can have a large number of functions that map from images to a (say) categorical label. In one case, this function could map the foreground object in the image to its label (e.g. the species of the cat). But in another case, this function could map the background object to its attribute (e.g. the color of the sofa). The "true" ImageNet label is but one possible function, and there could in principle be many more. Our approach can be used to represent an entire function, i.e. the function that maps from the image to the species of cat is assigned a different embedding than one that maps from the image to the color of the sofa.
>
> >(1) If this is few-shot learning, then it would make sense to compare to other few-shot methods, e.g. ones using meta-learning.
>
> We do in fact provide comparisons with the methods which perform few-shot learning for function data such as NP and CNP (Fig 2a, Table 1, Fig 3a,3b).
>
> >(2) If this is an extension of NP and CNP, then this method has no grounding probabilistically. It cannot provide estimates such as P(x|y), for example, which is the point of these methods. It is not very interesting that non-generative methods can outperform NP/CNP which are generative models.
>
> Our proposed method is not an extension of NP and CNP. We merely adopt their problem setting of regression functions and compare with them in that space. Moreover, our method can be used to obtain a generative model (decoder), which then performs comparable or outperforms that of NP/CNP even when the latter is trained end-to-end.
>
> Finally, CNP is not a probabilistic model but a deterministic one which is the closest baseline to our proposed framework.
>
> >(3) If this is learning representations for down-stream tasks, the the down-stream-focus of the experiment setup violates the "few-shot" part since only the original formulation (e.g. image inpainting) is few-shot. For down-stream tasks (e.g. image classification) this is rather a "sparse input" problem.
>
> We interpret the sparse input problem as a few-shot problem. However, our primary objective is to demonstrate that the representations we learn contain meaningful information that can be used for several downstream tasks such as classifying the content.
>
> > Despite the method being about learning representations, it does not compare to regular methods that treat images as images, not a set of (location,value) tuples. For example, a regular CNN may do better than the results reported in Figure 5.
>
> The setting of treating images as functions and treating them as tuples of (location, value) is only specific to the experiments in Sec 4.2. Figure 5. corresponds to scenes data in Sec 4.3. We treat scenes as deterministic functions that map camera viewpoints $x_i$ to images $y_i$ taken from those viewpoints. In this experiment, we indeed use CNNs to encode images, and the corresponding architecture is shown in Figure 12.
>
> > (4) If this is a method that can do both few-shot and learning representations, it should be still compared to regular few-shot and representation learning methods, especially because there are so many papers that do few-shot and representation learning, independently or combined.
>
> We do representations learning for functions and evaluate its usefulness by using it for multiple few-shot downstream tasks. We likewise compare this few-shot task performance with other few-shot methods defined for functions (CNP and NP). Since our method is specific to functions (treating samples as tuples of input-output pairs), we only compare with methods that operate in similar settings.

---

> > ### Author Response · Authors · 2020-11-17
> > **First response to R3 (part 2)**
> >
> > > The statistical significance is confusing. For example, Table 1 has 0.0087±0.0007 and 0.0078±0.0004 being significantly different with 3 runs, which does not sound plausible.
> >
> > 0.0007 and 0.0004 are the standard deviations over 3 runs. The standard errors (i.e. the standard deviation of the means) over 3 runs are 0.0004 and 0.0002 respectively. The corresponding p-value is 0.1253 under the null-hypothesis that both methods (NP and FCRL) perform equally well.
> >
> > > The method assumes each task is different. If there are many duplicated tasks (e.g. very similar images) then making each task representation different from all others would be suboptimal.
> >
> > We would expect this algorithm to yield an encoder that generates a latent space with a meaningful topology. We would expect similar functions to have representations close to each other for the following two reasons: 1) from a finite sample, we might not always be able to distinguish two similar functions, hence to avoid a potentially large loss in some ambiguous cases, it is better to map such functions to lie close to each other. 2) The encoder is a neural network that has a bias towards smooth functions. Hence it will prefer to map similar samples to similar latent representations. Our experiments confirmed these intuitions, where a linear probe on the representation can predict function properties accurately. Moreover, you can also this structure in the 2D latent mappings for scene images in Figure 13. The images with the same object properties are mapped closer in the latent space.
> >
> > > In eq. (5), it seems the input for downstream tasks include the pixels/view-images, so it will be unclear if the knowledge is from $X$ or $h_{\Phi}(D^T)$
> >
> > In Eq. 5, $X$ is the unlabelled data for downstream prediction tasks, and $Y$ is the corresponding prediction. The data $D^T$ corresponds to the observed context. We realize that this is not clear, and we will make it clear in the updated version.
> >
> > In conclusion, we hope to have resolved the concerns regarding the goals and motivations of the method. If you have further questions about the paper or our response, please feel invited to leave us a comment!

---

> > > ### Comment · AnonReviewer3 · 2020-11-25
> > > **Further comments**
> > >
> > > > The corresponding p-value is 0.1253
> > >
> > > That is larger than 0.05.
> > >
> > > > The images with the same object properties are mapped closer in the latent space.
> > >
> > > I agree it is not a problem in your experiments, but if someone were to take your method and apply in the real world, some functions are going to be extremely similar or even identical, and it may not fare well for them.

---

> > ### Comment · AnonReviewer3 · 2020-11-25
> > **Comments on the implication of the reply**
> >
> > Thank you for the clarification! The authors seems to choose (3) for the primary contribution and (1,4) as secondary. In that case, that is a bad title to pick, since you're primarily trying to learn a representation of the function. And the primary goal of the paper has nothing to do with few-shot learning despite the emphasis during motivation, since it is rather a sparse feature problem. The few-shot aspect is on a sub-problem of your primary problem and outside of your primary contribution.
> >
> > > The setting of treating images as functions and treating them as tuples of (location, value) is only specific to the experiments in Sec 4.2.  Figure 5 ... indeed use CNNs to encode images
> >
> > What I mean is, in Sec 4.2 you should have compared to methods using CNNs to encode images. In Figure 5, by analogy, you should compare to methods that encodes arbitrary number of images at once (a straightforward baseline would be to train a CNN that operates on individual images but averages their features at the end).

---

> ### Author Response · Authors · 2020-11-21
> **Second response to R3.**
>
> Hi Reviewer 3, we thank you again for your review. We now enlist our contributions at the end of the introduction to highlight the main problem we are tackling. We have expanded Sec. 2 to explicitly explain our goals and provide a few-shot interpretation of the problem setting concisely. For further clarity, we have added subsection 2.3 to explain the role of contrastively learned representations in functions learning space. We also accommodated your minor comments for Sec 3.1. If you still have some questions for us, then do let us know!

---

### Official Review · AnonReviewer4 · 2020-10-29
**An interesting idea of using contrast learning to distinguish different mapping functions, but there are still some major issues.**

**Rating:** 5
**Confidence:** 3

**Review:**

The paper works on the problem of few-shot learning. They propose to use contrastive learning to obtain good representations for different mapping functions. Experiments on 1-D, 2-D image and scene representation show certain advantages.

[Strengths]
- The idea of using contrastive learning to obtain good representations for different mapping functions is interesting and novel.
- The writing is in general clear and there are also a lot of experimental results.

[Weakness]
- There is a strong assumption that the mapping function during the few-shot task is one of the functions during training. Is this practical?
- The contrast learning only encourages those function representations to be different and it might not be able to learn the underlying function characteristics. Why can it help during down-stream few-shot tasks? In other words, I could use one-hot vector to represent each function. I could give the function index to the down-stream task. Is it helpful?
- Although contrast learning is hot, why must use contrast learning to learn function representation? You know the function labels, so you could also use triplet loss during the representation learning.
- Although the authors consider quite a few downstream tasks, all the comparisons are quite preliminary. It would be more convincing to show that in at least one task your approach can outperform SOTA or your function representation can improve SOTA.
- Figure 3(a) shows a better result with 50 shots, but with 200 shots. Why?
- There is no explanation for Fig. 2(b). It seems you use N samples to identify the function and the rest N-M samples to train the decoder. Is it correct? What's a good N here? What is the "S" operation?
- Eq. (4) does not look correct.
- Is Section 3.1 complete or not ? The organization is a bit wired.

---
Update:
- I think the idea of this paper is very interesting. The authors' responses also address most of my concerns. The remaining issues are: 1) the writing needs to be improved to make the paper easy to read; 2) I am still not so convinced with the experiments. It would be more convincing if the authors can demonstrate this on few-shot image classification to improve SOTA, for which the joint (x,y) distribution might be too big at the image level. On the other hand, I am not so familiar with this topic. If such experiments are quite normal in this area, then I am fine to increase the rating.

---

> ### Author Response · Authors · 2020-11-17
> **First response to R4 (part 1)**
>
> Hi Reviewer 4, thank you for reviewing our paper. We are glad that you find our idea of using contrastive learning to obtain representations of functions interesting and novel. Your questions are helping us improve the presentation of the paper, and we will upload a revision in the coming days. Please find a detailed response to your review below.
>
> > There is a strong assumption that the mapping function during the few-shot task is one of the functions during training. Is this practical?
>
> We only assume that the distribution over mapping functions is the same during representation-learning as it is in the downstream tasks -- we do not assume that the exact mapping function was encountered during training. This resembles the classical risk minimization setting where the test samples are drawn from the same distribution as the training samples (but the samples themselves are not identical). In this context, our problem setting is indeed practical and has been explored in other works, e.g. Conditional Neural Processes and Generative Query Networks.
>
> We believe that out-of-distribution generalization to mapping functions originating from a variety of different distributions is an interesting avenue of future research, but it is beyond the scope of the current paper.
>
> > The contrast learning only encourages those function representations to be different and it might not be able to learn the underlying function characteristics. Why can it help during down-stream few-shot tasks? In other words, I could use one-hot vector to represent each function. I could give the function index to the down-stream task. Is it helpful?
>
> This is a good question and indirectly asks why self-supervised contrastive learning should learn some meaningful representation of the functions instead of learning some trivial differences. We would expect this algorithm to yield an encoder that generates a latent space with a meaningful topology. We would expect similar functions to have representations close to each other for the following two reasons: 1) from a finite sample, we might not always be able to distinguish two similar functions, hence to avoid a potential large loss in some ambiguous cases, it is better to map such functions to lie close to each other in the latent space. 2) The encoder is a neural network that has a bias towards smooth functions. Hence it will prefer to map similar samples to similar latent representations. Our experiments confirmed these intuitions, where even a linear probe on the representation can predict function properties accurately.
> If the learned representations are trivial in the sense that each function is assigned an individual index, then we can not expect the linear decoder to achieve good accuracy on those downstream tasks.
>
> Moreover, we can also see this structure in the 2D latent mappings for scene images in Figure 13, where the images with the same object properties are mapped closer in the latent space.
>
> > Although contrast learning is hot, why must use contrast learning to learn function representation? You know the function labels, so you could also use triplet loss during the representation learning.
>
> We use contrastive learning because we do not assume to know the function labels, but only if two sets of observations originate from the same function. If the function labels are known, then a simpler objective would be to maximize the likelihood of correct label prediction.
>
> In fact, the connection of contrastive learning to deep metric learning and triplet losses has been explored [1]. The infoNCE contrastive objective can be approximated as a multi-class extension of the triplet objective [1]. One benefit of contrastive objective over triplet loss is that it provides an option to sample more negative samples. It has been shown empirically that the increasing number of negative samples helps to learn a better representation [2,3]. In this paper, we built upon the successes of contrastive objective and explored its utility in learning function representations. Nevertheless, we acknowledge that the triplet objective can be used to learn such representations and would be an interesting future direction to try.
>
> > Although the authors consider quite a few downstream tasks, all the comparisons are quite preliminary. It would be more convincing to show that in at least one task your approach can outperform SOTA or your function representation can improve SOTA.
>
> In our work, we consider strong baselines concerning functions' representation learning. The problem of defining downstream tasks on such representations (inferred from a few shots) is relatively unexplored, and we are not aware of any method that does that convincingly.
>
> --------- References --------
>
> - [1] https://arxiv.org/abs/1907.13625
> - [2] https://arxiv.org/abs/2002.05709
> - [3] https://arxiv.org/abs/1906.00910

---

> > ### Author Response · Authors · 2020-11-17
> > **First response to R4 (part 2)**
> >
> > > Figure 3(a) shows a better result with 50 shots, but with 200 shots. Why?
> >
> > Upon closer evaluation of the 50 shots results in Fig 3a, you can observe that the predicted digits are mostly hallucinated (because of fewer context samples available). However, with 200 shots, the predictions are constrained to be closer to the ground truth because of more context availability. We evaluated the MSE distance between the predicted and the ground truth MNIST images in the table to elaborate this further. It can be seen that the predictions on 200-shot results have less error than 50 shots results. The reported values are the mean and standard deviation of three independent runs.
> > \
> >  -------------------------------------------------------------#
> > \
> > |                    CNP                   |  FCRL
> > \
> >  -------------------------------------------------------------#
> > \
> > 50-shots   | 0.049 +-0.0002  | 0.050 +- 0.0001
> > \
> > 200-shots | 0.037 +-0.0002  | 0.039 +- 0.0001
> > \
> > #-------------------------------------------------------------#
> >
> > > There is no explanation for Fig. 2(b).
> >
> > Figure 2b shows how conditional downstream decoder is designed for the experiments in Sec 4.1 and 4.2 (Few-shot regression and few-shot image completion). It highlights the few-shot decoding at test time where we condition on $N$ samples to get the aggregated representation from our pretrained encoder first. The operation $S$ signifies the aggregation. The decoder conditions on this representation and predicts the targets $y_i$ for $M-N$ unlabelled $x_i$. We agree that more explanation is needed here and will include it in the manuscript's updated version.
> >
> > > It seems you use N samples to identify the function and the rest N-M samples to train the decoder. Is it correct? What's a good N here? What is the "S" operation?
> >
> > Given the explanation above, this is not entirely correct. We use N+M samples to train the decoder as done in CNP and NP, albeit to train a decoder only. The value of N depends on the problem domain (depending on how many samples we have available). For 1D regression functions we use N=20, for 2D regression function we use N=200, for MPI3D scenes we use N=2 and RLScenes we use N=20.
> >
> > > Eq. (4) does not look correct.
> >
> > Eq. 4 is the central piece of the work. Could you please specify what seems wrong?
> >
> > > Is Section 3.1 complete or not ? The organization is a bit wired.
> >
> > Thanks for pointing this out. We are working on improving this section in general and organizing it better. This concludes our first response to your review. If you have any questions then please do not hesitate to get in touch with us!

---

> ### Author Response · Authors · 2020-11-21
> **Second response to R4.**
>
> Hi Reviewer 4, we thank you again for your review and questions. We have added subsection 2.3 regarding ‘Motivation and the Intuition’ to explain the role of contrastive learning. Following your suggestions, we have also streamlined Sec. 3.1 and expanded the experiment sections to include more details about the downstream problems. If you have more questions, then do let us know.

---

### Official Review · AnonReviewer2 · 2020-10-31
**This paper is slightly below the acceptance threshold.  The general approach is clear and the evaluations show good performance but the writing needs improvement and the novelty is lacking.**

**Rating:** 5
**Confidence:** 4

**Review:**

The authors propose a novel training approach that uses a self-supervised signal to learn whether two sets of data are generated from the same function.  The goal is to align representations of different samples generated from the same function for use in few shot learning.

Background section should be expanded to provide a better understanding of the contrastive learning space.  In general section 2 needs to be improved.  The problem is not clear in this section as the main problem of few-shot learning is not clearly set up as an intuitive result of the approach of aligning representations of different samples that stem from the same function.

The writing in Section 3 needs to be improved.  Particularly the "Encoder Training" section as it is a little hard to follow.  There are K functions but the observations from the K functions are split into J sets of size N/J.  Is this just having N total observations that stem from K functions where positive pairs are pairs of observations from the same function and negative pairs are from different functions?  Or is this a method to sample a set of partial observations that are used in Equation 2 to generate the representation?  I am assuming the latter but this is not clear.

In Equation 4, when J > 2 then doesn't this become a multi-label problem where there are multiple pairs of observations from the same function?  I am confused on this equation in how it will prevent different positive pairs from competing with eachother.

I am not sure what the main contribution here is.  The paper describes a method for using contrastive learning to align representations of observations from the same generator.  The learning approach is not novel but the applications are interesting and the idea of aligning representations of observations from the same generator is good.

recommendation and reasoning

This paper is slightly below the acceptance threshold.  The general approach is clear and the evaluations show good performance but the writing needs improvement and the novelty is lacking.

I have read the author responses but my initial review remains unchanged.  The paper needs to be greatly improved in terms of clarity and the responses from the authors help but I think it is better off being resubmitted to another conference with heavy revisions.

---

> ### Author Response · Authors · 2020-11-17
> **First response to R2 (part 1)**
>
> Hi Reviewer 2, thank you for your review. We are glad that you found the application of our proposed method interesting and the idea of aligning the representations from the same generator good. We appreciate your comment about the lack of clarity in Sec. 2 and Sec. 3. We are working on accomodating your feedback in the updated version of the manuscript; meanwhile please find the first response to your review below.
>
> > Background section should be expanded to provide a better understanding of the contrastive learning space. In general section 2 needs to be improved. The problem is not clear in this section as the main problem of few-shot learning is not clearly set up as an intuitive result of the approach of aligning representations of different samples that stem from the same function.
>
> Thank you for pointing this out, we are working on improving our explanation of the problem setting. Our goal is to learn the representation of a given function from a **few** observed pairs $O^f =\\{(x_i, y_i)\\}_{i=1}^N$ such that this representation can be used interchangeably for multiple downstream problems defined on the same function (without requiring retraining). The contemporary methods such as CNP, NP, and GQNs learn such representations by conditioning on $O^f$ while optimizing for the conditional likelihood of the target distribution $p(y^{T}|x^{T}, O^f)$, where $T^f =\\{(x_i^{T}, y^{T}_i)\\}$ and $T^f \cap O^f = \emptyset$. This task-specific training results in a representation that performs suboptimally for the downstream problems other than the training problem, which is modeling $p(y^{T}|x^{T}, O^f)$. For e.g. if the model is trained on image completion (where $x$ corresponds to pixel location and $y$ to pixel values), then it does not perform well on image classification (that entails matching the set $O^f$ to a class). In contrast to this approach, we propose a self-supervised method, which first learns to extract a useful representation of a function only from the observed data $O^f$, without requiring any specific target $T^f$. By conditioning on this contrastively learned representation, we show that a variety of downstream problems can be reliably solved (including modeling $p(y^{T}|x^{T}, O^f)$).
>
> $\textbf{Intuition:}$ The intuition behind aligning representations of different samples that stem from the same function is to learn the representations of functions that are invariant to the sampling of the partial views $O^f$. For e.g., in learning a scene representation from multiple views, a single view can have some objects occluded. Learning to align a representation of that view with the other views of the same scene (where the object is potentially not occluded) can help extract a representation of the object that is invariant to views. Similarly, in 1D sinusoid functions, the learned representation should correspond to the sine wave's underlying parameters (the amplitude and the phase shift) but is invariant to where the sine wave is sampled. This paper seeks to learn such invariant representations of functions from only a few samples $O^f$. We will add these details in the manuscript.
>
> > There are K functions but the observations from the K functions are split into J sets of size N/J. Is this just having N total observations that stem from K functions where positive pairs are pairs of observations from the same function and negative pairs are from different functions? Or is this a method to sample a set of partial observations that are used in Equation 2 to generate the representation? I am assuming the latter but this is not clear.
>
> Thanks for pointing this out; we will improve this in the updated revision. For the sake of clarity, we provide detailed steps in Algorithm 1. In the following, we explain it in order.
> 1. We are provided with the observations from $K$ functions $O^{1:K}$.
> 2. Each observation $O^k$ contains $N$ examples $O^k = \\{O_n^k\\}_{n=1}^N$.
> 3. For setting up the contrastive objective, we split the observation set $O^k$ of size $N$ into $J$ subsets. Thus the size of resulting disjoints subsets is $N/J$.
> 4. We then apply the contrastive objective defined in Eq. 4.
>
> > In Equation 4, when J > 2 then doesn't this become a multi-label problem where there are multiple pairs of observations from the same function? I am confused on this equation in how it will prevent different positive pairs from competing with each other.
>
> This is a very good question. We avoid this issue by considering the natural setting of the contrastive objective of dealing with two positive views at a time i.e. for each view, we have only one positive match/label while computing the contrastive loss. We then compute the contrastive loss for all the pairwise combinations of the positive views and then sum them up: For instance, when J=3, we have 3C2 = 3 combinations of positive views. The term $\sum_{1\le i<j\le J}$ in Eq. 4 refers to that summation over all the combinations.

---

> > ### Author Response · Authors · 2020-11-17
> > **First response to R2 (part 2)**
> >
> > >I am not sure what the main contribution here is. The paper describes a method for using contrastive learning to align representations of observations from the same generator. The learning approach is not novel but the applications are interesting and the idea of aligning representations of observations from the same generator is good.
> >
> > As you mention, the key contribution is the insight that the good representation of a function can be learned by aligning the samples' representations that stem from the same underlying generator. This perspective on learning functions' representations is very different from the typical regression objectives used for learning such functions and is entirely new to our knowledge. With multiple, diverse experiments we show that such representations are not only robust to noise in inputs but also transfer well to the multiple downstream problems. Also, as recognized by R1, the use of contrastive objective in this setting is new, which is different from aligning the sample-wise representations.
> >
> > This concludes our first response to your review. If any aspect of it is unclear, please do not hesitate to get in touch with us!

---

> ### Author Response · Authors · 2020-11-21
> **Second response to R2.**
>
> Hi Reviewer 2, we thank you again for your review and questions. We have expanded Sec. 2 to explain our goals and provide a few-shot interpretation concisely. We have enlisted our main contributions at the end of the introduction and added subsection 2.3 to clearly define the motivation and the intuition behind the proposed framework. Following your suggestions, we have streamlined Sec. 3 and added details for the case when J >2. Feel free to get in touch with us if you have further questions.

---

### Official Review · AnonReviewer1 · 2020-10-31
**A few concerns about the paper.**

**Rating:** 5
**Confidence:** 4

**Review:**

The paper proposes to find a good representation of the underlying data generating function (data distribution) via contrastive learning. In contrast to existing works on applying contrastive learning to learn sample-wise representation, this idea is novel.

Pros:
The insight that "two sets of examples of the same function should have similar latent representations, while the representations of different functions should be easily distinguishable" is interesting. This can also relate to learning from a sets of tasks in meta-learning, where most tasks are related but some tasks are out of the distribution.

Cons:

On page 3, the illustration on training is like meta-learning. Is it? If so, please rewrite it using meta-learning notations. If not, please describe the difference with meta-learning.

Also, sentences under equation 2 use the terms "projected representations" and "encoded representations", what do they refer to? Can you use notations?

The paper provides a number of experiments on diverse data sets. These are valuable. However, as most of these tasks (especially Sec 4.2 and 4.3) are formulated into few-shot problems by authors, can you provide mathematical problem formulation at the beginning? It is hard to get why it is few-shot and what is the target.

Overall, I find some clarifications are required. The current version is not clear enough to convey its importance.

---

> ### Author Response · Authors · 2020-11-17
> **First response to R1.**
>
> We thank you for taking the time out to review our work. We are glad that you find our approach of learning the representations of data generating processes interesting and the use of contrastive learning in this setting novel. We also appreciate your feedback on improving the paper's presentation, especially regarding the part concerning the problem statement. While we are working on accommodating your suggestions in the manuscript, please find the first response to your review below.
>
> > On page 3, the illustration on training is like meta-learning. Is it? If so, please rewrite it using meta-learning notations. If not, please describe the difference with meta-learning.
>
> If you are referring to the main illustration on page 2, then it indeed portrays the training step for meta-learning function's representations. We used the notation similar to the one used in Conditional Neural Processes (CNP) as they explicitly study meta-learning for functions' representations. We acknowledge that a better explanation regarding the notation in section 2 would improve the readability. We are updating it in the revised manuscript.
>
> > Also, sentences under equation 2 use the terms "projected representations" and "encoded representations", what do they refer to? Can you use notations?
> 1. We first sample $N$ context points of a function $O^f = \\{(x_i, y_i)\\}_{i=1}^N$.
> 2. The encoded representations are obtained by average-pooling the pointwise transformations of the points i.e. $r = \frac{1}{N}\sum_{i=1}^{N}h_{\Phi}(x_i, y_i)$, where $h_{\Phi}(.)$ is the encoder network.
> 3. The projected representations are then the non-linear projections of these encoded representations i.e. $g_{(\phi, \Phi)}(O^f) = \rho_{\phi}(r)$, where $\rho_{\phi}$ is an MLP with one hidden layer.
> We will add these steps explicitly in the updated manuscript.
>
> > The paper provides a number of experiments on diverse data sets. These are valuable. However, as most of these tasks (especially Sec 4.2 and 4.3) are formulated into few-shot problems by authors, can you provide mathematical problem formulation at the beginning? It is hard to get why it is a few-shot and what is the target.
>
> For each task, we consider a set of $\textbf{few}$ input-output pairs $O^f$ as the context dataset or the partial observation of a function. Using our pre-trained encoder $h_{\Phi}(.)$, we first infer the encoded representation $r$ and train a different conditional decoder $p_{\psi}(.|r)$ for each downstream problem (while keeping $h_{\Phi}(.)$ fixed). Depending on the nature of the downstream problem, the targets differ. We define different conditional distributions (and their corresponding likelihood objectives) as follows:
>
> $\textbf{Sec 4.2a (Few-shot Image Completion):}$ Here, the downstream problem is a regression, and we consider the same setting as CNP and NP in all aspects. In addition to the $N$ context points, a set of $M$ unlabelled target points $T = \\{(x_i^T,y_i^T)\\}_{i=N}^{N+M}$ are sampled.  Here, N is sampled randomly from the range [2, 200] and $M$ from [0, 200-N].
>
> The conditional distribution is defined as $P(y^T|x^T, O^f)$, and the decoder $p_{\psi}$ is trained to minimize the following negative conditional log-likelihood as $\min_{\psi}$ $\mathop{\mathbb{E}_{f \sim p(f)}}$ $[\mathop{\mathbb{E}}$ _{N+M} $[\log p\_\psi(y_i^{N+M} | x_i^{N+M}, r)]]$
>
> $\textbf{Note:}$ For training purposes, following CNP and NP, we also use $N$ observed context points in $T$ i.e. use $(N+M)$ datapoints for optimizing the conditional likelihood objective, albeit to train only the decoder $p_\psi$.
>
> $\textbf{Sec 4.2b (Few-shot Content Classification):}$ Here, the downstream problem is classification and the goal is to train a linear decoder $p_{\psi}$ to identify the digit class by observing only $N$ pixels ($x_i$ is pixel location and $y_i$ is pixel value). The target is the one-hot vector, specifying the MNIST class label of each function i.e. $T = {y_{onehotLabel}^f}$. The conditional distribution is then $P(T|O^f)$ and the conditional likelihood objective is $\min_{\psi}$ $\mathop{\mathbb{E}_{f \sim p(f)}}$ $[\log p_\psi(y_{onehotLabel}^f|r)]$.
>
> $\textbf{Sec 4.3 (Scenes as Functions):}$ This setting of inferring a scene representation from only limited tuples of camera viewpoints and the corresponding views is considered in Generative Query Networks, where $x$ are viewpoints, and $y$ are images taken from that viewpoint. We devise few-shot downstream problems to investigate how informative and useful the inferred scene representations are. 1) The first task is to train a linear decoder $p_\psi$ to infer one-hot labels corresponding to the factors of variations in a scene. 2) The second task is to train an MLP policy $p_\psi$ to maximize the reward function of an RL agent's manipulation task while conditioning on the inferred encoded representation $r$ only.
>
> This concludes our first response to your review. If any questions remain, please feel invited to interact with us!

---

> ### Author Response · Authors · 2020-11-21
> **Second response to R1.**
>
> Hi Reviewer 1, we thank you again for your review. Following your suggestions, we have simplified the notation and presented it clearly in Sec. 2. We have also added more details to Sec. 3 and added mathematical formulations of each few-shot downstream problem in Sec 4. In case you have more questions, then feel free to contact us.

---

### Author Response · Authors · 2020-11-21
**Summary of the updates.**

We thank all the reviewers for their valuable feedback on improving the quality of this work. We have updated the paper accordingly and summarise the main changes below.

- We explicitly enlist our main contributions at the end of the introduction.
- We have expanded Sec. 2.1 of ‘Problem Setting.’ It now clearly explains the goals and the few-shot setting of the proposed framework.
- We have added Sec. 2.3 of ‘Motivation and Intuition’ to add more clarity.
- We have streamlined Sec. 3 by considering the individual suggestions made by R1, R2, and R3.
- We have added more details to the experiments in Sec. 4 and explain the downstream tasks (with the associated objectives) in their specific subsections.

---

### Decision · Program_Chairs · 2021-01-07
**Final Decision**

**Decision:**

Reject

**Comment:**

While the authors provided extensive responses to the reviewers and most of the reviewers did a good job of accounting for the author responses the final ratings for this paper was unanimously 5s -- all marginally below acceptance. The paper's positioning, writing were identified as key issues that remained to be addressed. The AC recommends rejection.